# Structural and dynamic mechanisms for coupled folding and tRNA recognition of a translational T-box riboswitch

Xiaolin Niu [1], Zhonghe Xu [1], Yufan Zhang[2], Xiaobing Zuo [3], Chunlai Chen [1,4] ✉ & Xianyang Fang [1,2] ✉

T-box riboswitches are unique riboregulators where gene regulation is mediated through interactions between two highly structured RNAs. Despite extensive structural insights, how RNA-RNA interactions drive the folding and structural transitions of T-box to achieve functional conformations remains unclear. Here, by combining SAXS, single-molecule FRET and computational modeling, we elaborate the folding energy landscape of a translational T-box aptamer consisting of stems I, II and IIA/B, which $Mg^{2+}$-induced global folding and tRNA binding are cooperatively coupled. smFRET measurements reveal that high $Mg^{2+}$ stabilizes IIA/B and its stacking on II, which drives the pre-docking of I and II into a competent conformation, subsequent tRNA binding promotes docking of I and II to form a high-affinity tRNA binding groove, of which the essentiality of IIA/B and S-turn in II is substantiated with mutational analysis. We highlight a delicate balance among $Mg^{2+}$, the intra- and inter-molecular RNA-RNA interactions in modulating RNA folding and function.

RNAs play essential roles in multiple cellular processes including gene expression, processing, catalysis and gene regulation[1]. The functional diversity of RNAs is determined by their propensity to fold into distinct secondary and tertiary structures, interact with other molecules to form complex quaternary assemblies, and transit dynamically between alternative conformational states in response to specific cellular signals[2]. Previous studies have provided important insights into how RNA interacts with ion, metabolite or protein to guide the folding and conformational dynamics of RNAs to facilitate cellular functions[3–6]. However, our understanding of how RNA-RNA interactions occur and drive the RNA folding and conformational dynamics remains relatively limited.

The T-box riboswitches are unique RNA-based regulators, in which gene regulation is accomplished through interactions between two highly structured noncoding RNAs, namely the T-box and a tRNA ligand. Mainly found in Gram-positive bacteria, T-box riboswitches are structured cis-regulatory RNA elements located in the 5' untranslated regions (UTR) of mRNAs encoding aminoacyl tRNA synthetase and proteins involved in amino acid synthesis and transport[7]. Canonical T-box riboswitches contain highly conserved helical structures designated as stems I, II, IIA/B and III that are upstream to an anti-terminator (transcriptional T-box) or anti-sequestrator (translational T-box) element (Fig. 1a). These elements can be further composed into an aptamer domain (stems I, II and IIA/B) responsible for specific binding of tRNA and a downstream discriminator domain (stem III and anti-terminator/anti-sequestrator elements) which senses the tRNA aminoacylation state and executes conformational switching to attenuate transcription or initiate translation[8–10]. Due to their modular structure, high conservation among different species and lack of human homologs, T-box riboswitches have been identified as potential drug targets to develop RNA-targeted antibiotics[11–15].

Recently, several high-resolution structures for T-box riboswitches in complex with their cognate tRNAs have been obtained[16–20], revealing a wealth of complex RNA-RNA interactions that create

[1]Beijing Frontier Research Center for Biological Structure, School of Life Sciences, Tsinghua University, Beijing 100084, China. [2]Key Laboratory of RNA Science and Engineering, Institute of Biophysics, Chinese Academy of Sciences, Beijing 100101, China. [3]X-ray Science Division, Argonne National Laboratory, Lemont, IL 60439, USA. [4]State Key Laboratory of Membrane Biology, Tsinghua University, Beijing 100084, China. ✉e-mail: chunlai@mail.tsinghua.edu.cn; fangxy@ibp.ac.cn

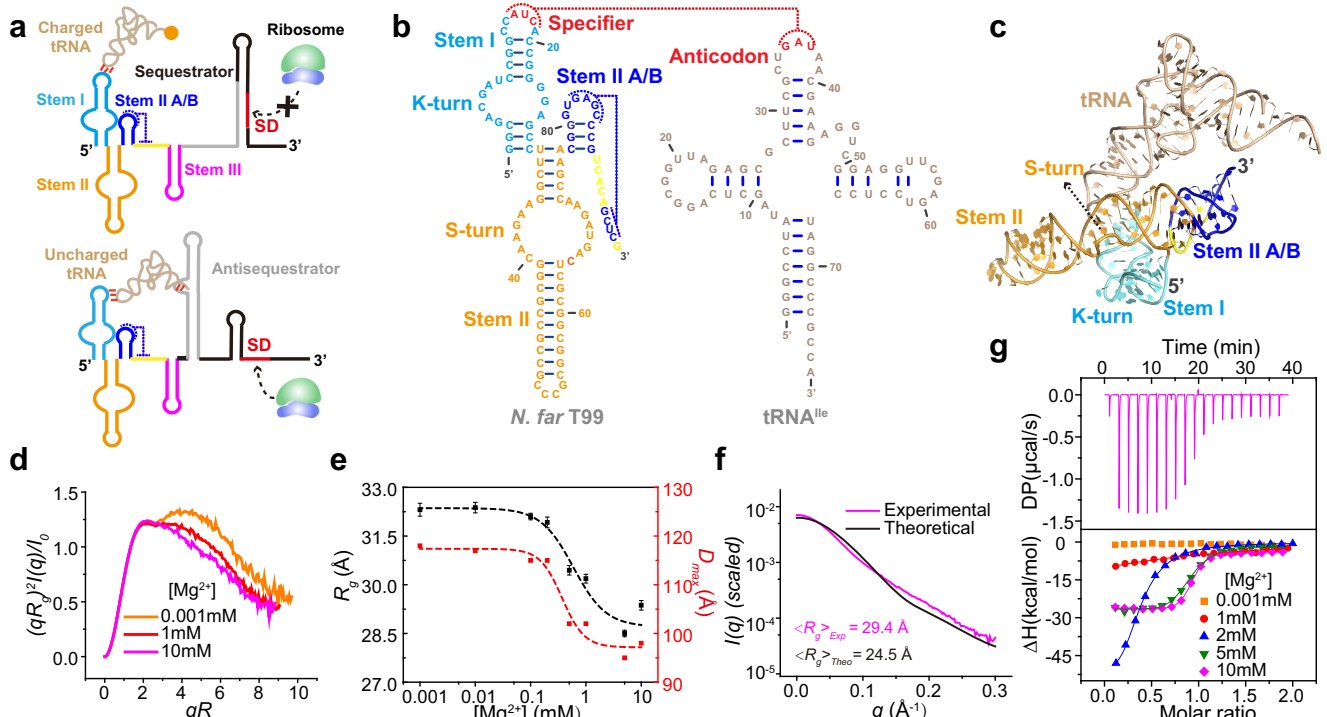

**Fig. 1 | Structure of T99 in *apo* form probed by SAXS. a** The proposed translational regulation mechanism by T-box riboswitch. **b** Sequence and secondary structure of the *N. farcinica ileS* T-box and tRNA^Ile used in this study. Red line indicates the interaction between the T-box Specifier loop and the tRNA anticodon. **c** Crystal structure of the *ileS* T-box-tRNA^Ile complex colored as in **b** (PDB id: 6UFM). **d** The dimensionless Kratky plots for T99 in different Mg²⁺ concentrations. **e** Plots of $R_g$ (black) and $D_{max}$ (red) as a function of Mg²⁺ for T99. Each data point represents an independent experiment ($n=1$) and the error bars are propagated uncertainties calculated by GNOM. **f** The theoretical scattering curve (shown as black line) calculated from the X-ray crystal structure in **c** was fitted with experimental scattering curves (shown as purple line) for T99 in 10 mM Mg²⁺. **g** ITC analysis for tRNA (~200 μM) binding by T99 (~20 μM) in different Mg²⁺ concentrations. Source data for panels **d**–**g** are provided as a Source Data file.

high-affinity tRNA binding sites by T-boxes. Multiple contact sites have been revealed between the T-boxes and the tRNAs, which are driven and stabilized by a large set of intra- and inter-molecular RNA-RNA interactions. For example, in the crystal structure of a translational *ileS* T-box in complex with tRNA[16,19], the inter-molecular base-pairing interactions between the tRNA^Ile anticodon and the Specifier are laterally reinforced by the intra-molecular interactions between stem I containing a K-turn and stem II bearing an S-turn, which are arranged perpendicularly and joined at their bases near the stem IIA/B pseudoknot. The pseudoknot is coaxially stacked with stem II to extend the dsRNA trajectory and anchors the base of the stem I to dock with the middle section of stem II, creating an extended binding groove for the tRNA binding (Fig. 1b, c). While intra-molecular RNA-RNA interactions occurring are generally essential for RNA tertiary folding, inter-molecular RNA-RNA interactions between different RNA molecules are important to form larger RNA assemblies. Therefore, T-box riboswitch represents an excellent paradigm to understand RNA-RNA interactions.

RNAs are inherently dynamic and flexible molecules that sample a large ensemble of conformations in solution. As the high-resolution structures of tRNA-bound T-boxes only provide a static snapshot of a single conformation, structural information about the tRNA-free T-boxes is lacking. As a result, it remains largely unknown how the complex RNA-RNA interactions drive the alternative T-box folding pathways, the kinetics of ligand binding and ligand-driven conformational changes leading to gene regulation. A deeper understanding of these topics is essential to comprehend the T-box regulatory mechanisms at the molecular level. Recently, various biophysical methods including small-angle X-ray scattering (SAXS) and single-molecule fluorescence resonance energy transfer (smFRET) have been

applied to investigate the structural dynamics and kinetics of a transcriptional *glyQS* T-box binding to tRNA[21–23], unraveling a two-step hierarchical tRNA sensing mechanism and substantial conformational changes upon tRNA binding. However, due to their large structural differences, little is known about the folding pathway and structural dynamics of any translational T-box riboswitch.

In this study, we have combined isothermal titration calorimetry (ITC), SAXS with smFRET techniques to probe the folding and conformational dynamics of the aptamer domain (T99) of a translational *ileS* T-box riboswitch from *Nocardia farcinica* (Fig. 1b). Our results show that the global folding and high-affinity tRNA binding of T99 are cooperatively coupled and highly depend on Mg²⁺. To enable in-detail 3D mapping of the conformational changes of T99 induced by Mg²⁺ and tRNA binding by smFRET, we develop a site-specific dual fluorophore labeling scheme based on the NaM-TPT3 unnatural base pair (UBP) system, which allows for a comprehensive labeling scheme including 7 pairs of fluorophore labels across the T99. smFRET analysis reveals that Mg²⁺ has a minimal impact on the folding of individual stems I and II, but is required for the formation of stem IIA/B. Consequently, the Mg²⁺-facilitated inter-stem preorganization between stems I and II to form a competent tRNA binding conformation is dynamically coupled with the formation of stem IIA/B pseudoknot and its coaxial stacking with stem II. Subsequent binding of the cognate tRNA^Ile anticodon by stem I Specifier induces the docking of stem I on the S-turn and surrounding regions of stem II, and concomitantly, moving stem IIA/B away, which further stabilizes the Specifier-anticodon interaction. The essentiality of stem IIA/B and stem II S-turn regions in such processes is further demonstrated by dynamic analysis and 3D structural visualization of several point mutants and transcriptional intermediates of T99. Taken together, our results

delineate a conformational energy landscape for a translational T-box riboswitch in which cooperative RNA-RNA interactions drive RNA folding and promote high-affinity tRNA recognition.

## Results

### Global conformational changes of T99 induced by $Mg^{2+}$ and tRNA binding

$Mg^{2+}$ ions are known to be important for the folding and ligand binding of many riboswitches[5,24,25]. To understand how $Mg^{2+}$ affects T99 RNA, we first probe its global folding and conformational changes through SAXS (Supplementary Fig. 1a). The dimensionless Kratky plot, plotted as $(qR_g)^2I(q)/I_O$ versus $qR_g$, reflects the degree of flexibility and compactness of the molecule in solution. At low $Mg^{2+}$, T99 displays a significant enrichment in the high scattering angles, characteristic of an unfolded conformation. As $Mg^{2+}$ increases, there are significant changes in the plots resulting in transitions from a bimodal peak to a more bell-shaped curve indicative of a folding event (Fig. 1d). The structural parameters derived from the PDDFs, including the radius of gyration ($R_g$) and the maximum particle dimension ($D_{max}$) were plotted as a function of $Mg^{2+}$ (Fig. 1e). Both $R_g$ and $D_{max}$ of T99 RNA decrease as $Mg^{2+}$ increases, consistent with a $Mg^{2+}$-induced structural transition from the unfolded to folded states. The theoretical scattering curve from the crystal structure of tRNA-bound T-box aptamer fits poorly with the experimental scattering profile of T99 alone at 10 mM $Mg^{2+}$ (Fig. 1f). Additionally, the predicted $R_g$ and $D_{max}$ value of tRNA-bound T99 was smaller than that of the T99 RNA alone at 10 mM $Mg^{2+}$, suggesting that tRNA-bound T99 RNA adopts a significantly different conformation from that *in apo* form. Taken together, $Mg^{2+}$ is essential for the proper folding of T99 and tRNA binding at high $Mg^{2+}$ induces further conformational changes in T99.

ITC measurements were also performed for the T99 in the presence of varying $Mg^{2+}$. Interestingly, the binding of $tRNA^{Ile}$ to T99 highly depends on $Mg^{2+}$ (Fig. 1g). While no tRNA binding can be observed in low $Mg^{2+}$, the binding affinities become higher as $Mg^{2+}$ increases (Supplementary Table 1). Intriguingly, the enthalpy change ($\Delta H$) in 2 mM $Mg^{2+}$ is significantly larger than that in higher $Mg^{2+}$, indicating that the binding properties for T99 in high $Mg^{2+}$ may differ from that in low $Mg^{2+}$ (discussed later). Given that $Mg^{2+}$-induced folding is essential for high-affinity tRNA binding and tRNA binding induces further structural changes in T99 RNA, these two processes are cooperatively coupled.

### Comprehensive fluorescent labeling scheme of T99 empowered by UBP system

Though SAXS is powerful in probing the global structure and conformational dynamics of biomolecules, due to its relatively low resolution, it hardly provides any detailed local information. In contrast, smFRET has emerged as a versatile tool to monitor the inter-domain or inter-subdomain conformational rearrangements in a biomolecule with high temporal and spatial resolution. The prerequisite for smFRET measurements is to achieve site-specific orthogonal labeling of the biomolecule with two spectrally overlapping dyes. While a variety of approaches have been developed to achieve such a goal, efficient labeling of large RNAs remains challenging[26]. Recently, we have achieved site-specific fluorescent labeling of large flaviviral RNAs up to 719 nts using the NaM-TPT3 UBP system[27,28] (Fig. 2a). However, such a strategy has been only used for single-site labeling.

To achieve site-specific dual fluorophore labeling using the UBP system, we synthesized an alkyne-modified rNaMTP derivative ($rNaM^{CO}$) which can be incorporated into RNA orthogonally with amine-modified rTPT3TP ($rTPT3^A$). Using the similar strategy[29], DNA templates containing an upstream T7 promoter, a dNaM and a dTPT3 modifications at specific sites are prepared by overlapping PCR first. Then the $rNaM^{CO}$ and $rTPT3^A$ are incorporated into specific sites of the RNA by in vitro transcription. The $rNaM^{CO}$- and $rTPT3^A$-modified transcripts are subsequently conjugated with azide-modified Cy5 and NHS ester-modified Cy3 dyes through click chemistry and amine-NHS ester reactions, respectively (Fig. 2b, c), resulting in an RNA labeled with a FRET dye pair.

Facilitated with this dual-labeling strategy, we designed 7 pairs of labeling sites located within the intra-stem or between the inter-stems of T99 (Fig. 2d), constituting a comprehensive distance network and thus allowing accurate 3D mapping of its structural dynamics. The sequences of each labeling construct were reported in Supplementary Fig. 2. Electrophoretic mobility shift assay (EMSA) showed that the fluorescent dye labeling has a minor effect on tRNA binding by T99 (Supplementary Fig. 3). To enable single-molecule imaging of RNAs over extended periods using a wide-field total internal reflection fluorescence microscopy (TIRFM) setup, the 3'-end of T99 was co-transcribed with a single-stranded RNA sequence to anneal with its complementary DNA oligomer containing a 5' biotin for surface immobilization (Fig. 2e). Using this approach, conformational dynamics for T99 under different conditions could be tracked over time by recording the emission intensities of both Cy3 and Cy5 fluorophores within hundreds of individual surface-immobilized molecules simultaneously. FRET efficiencies were used to assess time-dependent changes in distance between the fluorophore pairs.

### Intra-stem conformational dynamics of T99

To have an in-detail 3D mapping of its conformational changes induced by $Mg^{2+}$ and tRNA binding, we first characterize the dynamics of individual stem of T99 using smFRET (Fig. 3).

Stem I harbors a conserved kink-turn (K-turn) structural motif that bends the duplex RNA trajectory by ~120°, acting as a geometry tool to facilitate the docking between stems I and II (Fig. 1c). To probe dynamics in stem I, Cy3 and Cy5 were covalently linked to site G1 via the 5'-end labeling method, and site C14 at the apical loop of stem I using the UBP-based strategy, respectively. The time-dependent Cy3 and Cy5 intensities of the T99/1-14 construct were collected and recorded at three conditions, in the absence of $Mg^{2+}$, in the presence of high $Mg^{2+}$, and in the presence of high $Mg^{2+}$ and tRNA (Fig. 3b). The peak center and population of each state in the FRET histograms were extracted through the Gaussian fitting. At all conditions, T99/1-14 samples a middle- (~0.45) and a high-FRET (~0.7) states, revealing a dynamic stem I in solution (Fig. 3b, Supplementary Fig. 4). The distance between the labeling sites in the crystal structure is estimated to be about 37 Å, which should result in a high-FRET signal. Thus, the two states likely correspond to unkinked and kinked conformations, respectively. In the absence of $Mg^{2+}$, T99/1-14 mostly occupies the high-FRET state (~56%) and transits into middle-FRET occasionally. In 20 mM $Mg^{2+}$, the population occupying the high-FRET state substantially increased to 74%, however, further addition of tRNA slightly decreased the population at the high-FRET state (~65%). These findings are consistent with previous studies that the K-turn is largely formed even in the presence of only monovalent salt which can be further stabilized by $Mg^{2+}$ [30]. However, tRNA binding doesn't cause significant structural rearrangements in stem I.

Stem II comprises a long hairpin structure intercalated with a conserved S-turn motif, which is essential for tRNA recognition. To study the dynamics of stem II, A35 and G54 spanning the S-turn were chosen for fluorescent labeling (T99/35-54). Under all three conditions, T99/35-54 displayed predominantly a ~0.4 middle-FRET state. The FRET efficiency increased slightly in high $Mg^{2+}$, however, no further change was observed upon the addition of tRNA (Fig. 3c).

We next explore the folding of stem IIA/B which forms a compact pseudoknot in the crystal structure. To do so, we prepared a T99/82-99 construct. The smFRET data obtained for T99/82-99 were depicted in Fig. 3d, revealing a highly dynamic conformation that exchanges between middle- (~0.5) and high-FRET (~1.0) states, corresponding to unfolded and formed pseudoknots, respectively. Numerous studies

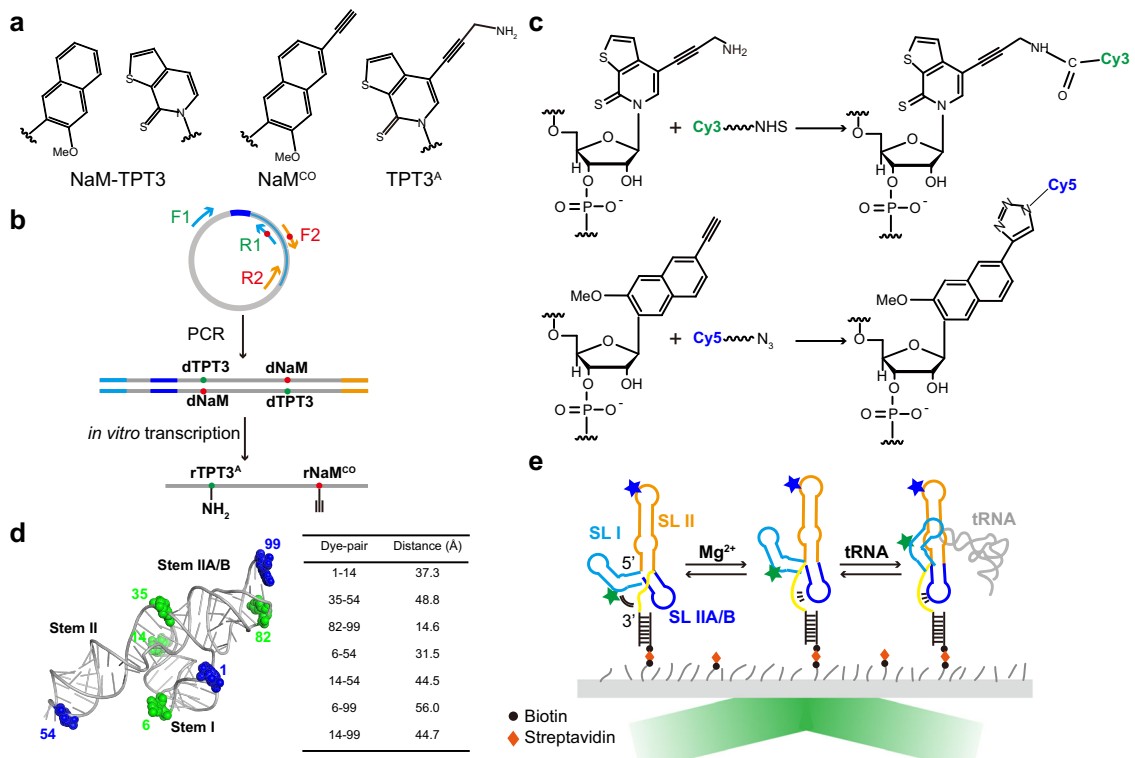

**Fig. 2 | The unnatural base pair-based labeling strategy facilitates TIRFM-based smFRET analysis of the folding of T99. a** Chemical structures of parental NaM-TPT3 unnatural base pair, alkyne-modified NaM (rNaM$^{CO}$) and amine-modified TPT3 (rTPT3$^A$). **b** DNA templates of T99 containing T7 promoter (shown in blue) and dNaM/dTPT3 modification at the template strand were prepared by two-step overlapping PCR, which was followed by in vitro transcription using rNTP mix supplemented with rTPT3$^A$TP and rNaM$^{CO}$TP allowing for site-specific modification of the transcripts with reactive amine and alkyne. **c** Conjugation of azide-modified

Cy5 dye and NHS-modified Cy3 dye with rNaM$^{CO}$ and rTPT3$^A$-modified RNA via click chemistry and NHS-amine reaction, respectively. **d** The homology 3D structure model of T99 was built up by ModeRNA[56]. Selected fluorophore labeling residues were shown as green and blue spheres. Distances between each pair of labeling position in the structure are shown in the table. **e** Schematic representation of the TIRFM-based smFRET analysis of the folding of T-box riboswitch. Cy3 and Cy5 were shown as green and blue stars, respectively.

---

have shown that Mg$^{2+}$ promotes and stabilizes the formation of pseudoknot[31,32]. In accord with this, the population of the high-FRET state in T99/82-99 increases substantially from ~42% in the absence of Mg$^{2+}$ to 57% in 20 mM Mg$^{2+}$. Further addition of tRNA only causes a modest increase (~3%) in the occupancy of high-FRET state. The Mg$^{2+}$ dependence of stem IIA/B pseudoknot formation was further verified in a homologous translational T-box riboswitch from *Mycobacterium tuberculosis*. Similar to T99/82-99, T87/71-3'end interconverts between a middle- (~0.5) and a high-FRET (~1.0) states under all three conditions (Fig. 3e). Addition of Mg$^{2+}$ prominently increases the fraction of high-FRET state from 23% to 52%, consistent with a stabilizing role of Mg$^{2+}$ in pseudoknot formation[32], which could be explained by the observation of a Mg$^{2+}$ binding site near the stem IIA/B region[16]. Similarly, further addition of tRNA only slightly increases the population of the high-FRET state.

### Inter-stem conformational dynamics of T99
Given that Mg$^{2+}$ and tRNA binding induce significant global conformational changes in T99 (Fig. 1d–f), large inter-stem motions are expected. As stem II is relatively rigid and stacked against stem IIA/B in the crystal structure, we next characterized the collective inter-stem motions between stems I and II and between stems I and IIA/B of T99 (Fig. 4a).

The labeling construct T99/6-54 was used to characterize the relative inter-stem motions between stems I and II. smFRET data for T99/6-54 were obtained under the same three conditions as above and summarized in Fig. 4b. In the absence of Mg$^{2+}$, a highly populated low-FRET (~0.08) state was observed, suggesting that the inter-dye

distance in T99/6-54 is too far (e.g., >70 Å). However, the distance between the labeling sites estimated from the crystal structure is about 31 Å, which is expected to give a high-FRET signal. Thus, stems I and II are likely to coaxially stack and adopt an undocked conformation at low Mg$^{2+}$. At high Mg$^{2+}$ which the T99 RNA is presumed to be folded and competent for high-affinity tRNA binding, the low-FRET state was shifted to a low-to-middle-FRET (~0.32) state, indicating that stem I is moving close to stem II. Upon addition of tRNA, a middle-FRET (0.57) state emerged and was highly populated (65%), presumably corresponding to the tRNA-bound conformation since T99/6-54 alone mainly stays in the low-to-middle-FRET state and rarely samples the middle-FRET state. These data indicate that while high Mg$^{2+}$ promotes the preorganization of stems I and II into a competent conformation, subsequent tRNA binding further induces their docking, resulting in a significant compact and more closed conformation between stems I and II. We thus defined the low-to-middle-FRET in high Mg$^{2+}$ and the middle-FRET in the presence of tRNA as predocked and docked states, respectively. To substantiate these findings, we also performed smFRET measurements for another labeling construct (T99/14-54). Similar phenomena were observed for T99/14-54 under the same conditions (Fig. 4c). While high Mg$^{2+}$ shifted a low-FRET (~0.16) state in the absence of Mg$^{2+}$ to a low-to-middle-FRET (~0.34) state, the addition of tRNA at high Mg$^{2+}$ results in the emergence of a pronounced middle-FRET (~0.63) state coexisting with the low-to-middle-FRET state.

To characterize the relative inter-stem motions between stems I and IIA/B, we synthesized the T99/6-99 construct. The inter-site distance estimated from the crystal structure is ~56 Å, which is expected

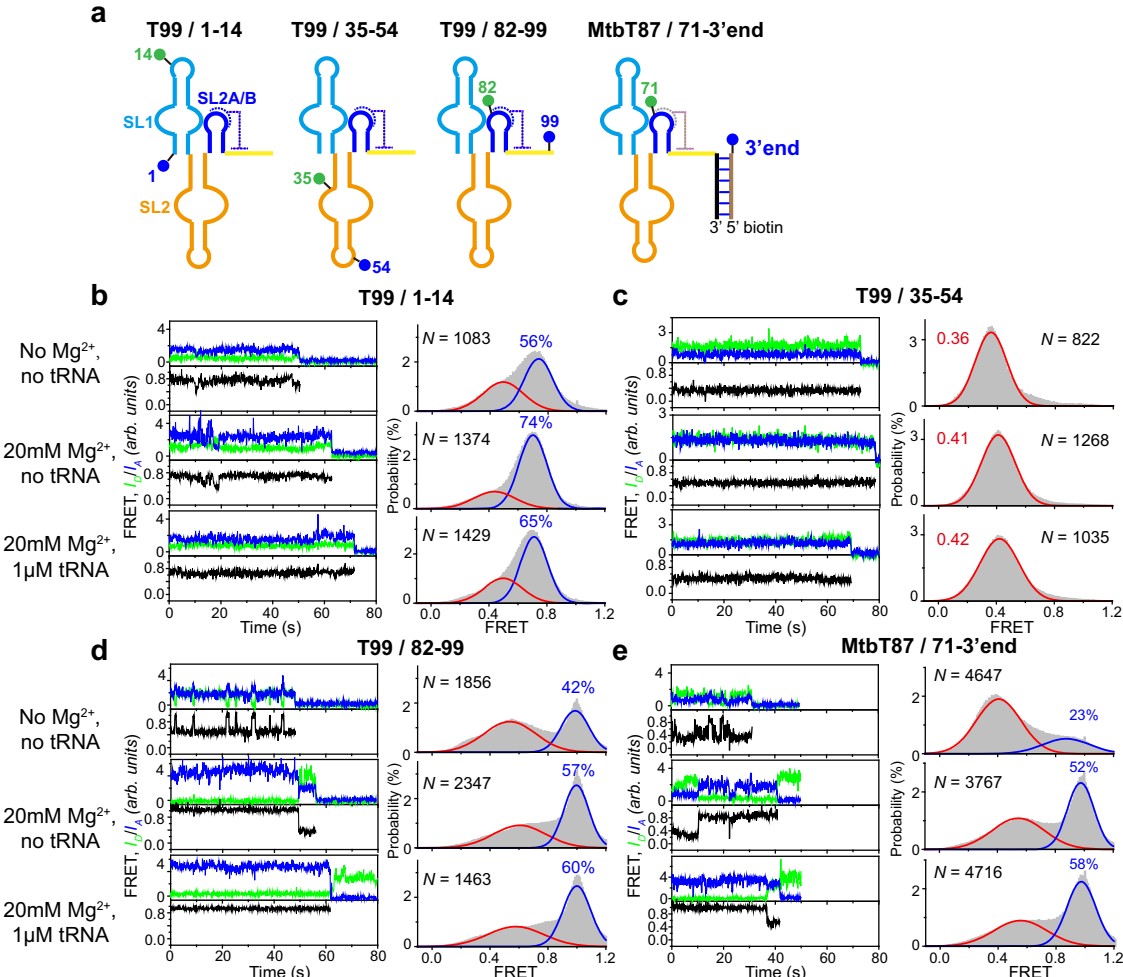

**Fig. 3 | Mg²⁺ and tRNA mediated intra-domain structural changes of T99 analyzed by smFRET. a** Schematic representation for the secondary structure of each smFRET construct. The blue and green circle indicates Cy5 and Cy3 fluorophore, respectively. **b–e** Representative smFRET traces (left) and FRET histograms (right) for each smFRET construct in **a** in three different conditions: No Mg²⁺, 20 mM Mg²⁺ and 20 mM Mg²⁺ in the presence of 1 μM tRNA. Green, blue and black lines represent the Cy3 intensity, Cy5 intensity and FRET efficiency, respectively. Histograms were well fitted with the Gaussian peaks, shown in red and blue for the low- and middle- or high-FRET states, respectively. *N* denotes the total number of traces to generate histograms from three independent experiments (*n* = 3). Source data for panels **b–e** are provided as a Source Data file.

to give a low-FRET signal. smFRET data for T99/6-99 under all three conditions were obtained and summarized in Fig. 4d. T99/6-99 in the absence of Mg²⁺ exhibited a major but broad FRET population, which is likely due to the intrinsic dynamics of stem IIA/B pseudoknot. Increasing Mg²⁺ to 20 mM shifted the peak of FRET distribution to a middle value (~0.49), indicating that stems I and IIA/B are moving towards each other. Interestingly, upon the addition of tRNA, a sharp and highly populated low-FRET (~0.26) state emerges and coexists with the middle-FRET state, suggesting that tRNA binding drives stem I moving away from stem IIA/B to dock with stem II. smFRET measurements were also performed for another construct (T99/14-99) which C14 and G99 were labeled with Cy3 and Cy5, respectively. Similar phenomena were observed for T99/14-99 as that for T99/6-99 (Fig. 4e). While high Mg²⁺ shifted a low-to-middle-FRET (~0.36) state in the absence of Mg²⁺ to a middle-FRET (~0.45) state, further addition of saturating tRNA at high Mg²⁺ results in the emergence of a pronounced low-FRET (~0.25) state.

Taken together, these smFRET data reveal a sequential folding and binding mechanism for T99 that high Mg²⁺ facilitates the folding of stem IIA/B pseudoknot and its stacking on stem II for preorganization of stems I and II to form a competent tRNA binding conformation, and subsequent tRNA binding drives further docking of stem I on stem II,

and concomitantly, moving away from stem IIA/B (Fig. 4f, Supplementary Movie 1 and Supplementary Movie 2).

## Stem IIA/B pseudoknot is essential for pre-docking of stems I and II

Stem IIA/B pseudoknot makes no direct contact with tRNA in the crystal structure, however, previous studies showed that disruption of stem IIA/B pseudoknot drastically reduced tRNA binding in vitro, suggesting that the structural integrity of IIA/B pseudoknot is essential for the function of T-box riboswitch[19]. It's speculated that stem IIA/B might serve as a geometry hub to organize the pre-docking of stems I and II for high-affinity tRNA binding.

To experimentally delineate the roles of the structural integrity of stem IIA/B in pre-docking of stems I and II, two single-point mutants (G85C/6-54 and U90C/6-54) that impair the IIA/B pseudoknot by disrupting two conserved base triple interactions and two deletion mutants that remove IIB (T89) or the complete IIA/B pseudoknot (T77) of T99/6-54 were constructed (Fig. 5a). The docking dynamics of stems I and II of these mutants alone in 7.5 mM Mg²⁺ were investigated (Fig. 5b). Under such condition, T99/6-54 is well folded and exhibits a narrow FRET distribution. While both G85C/6-54 and U90C/6-54 alone exhibited a much broader FRET distribution with a peak value smaller

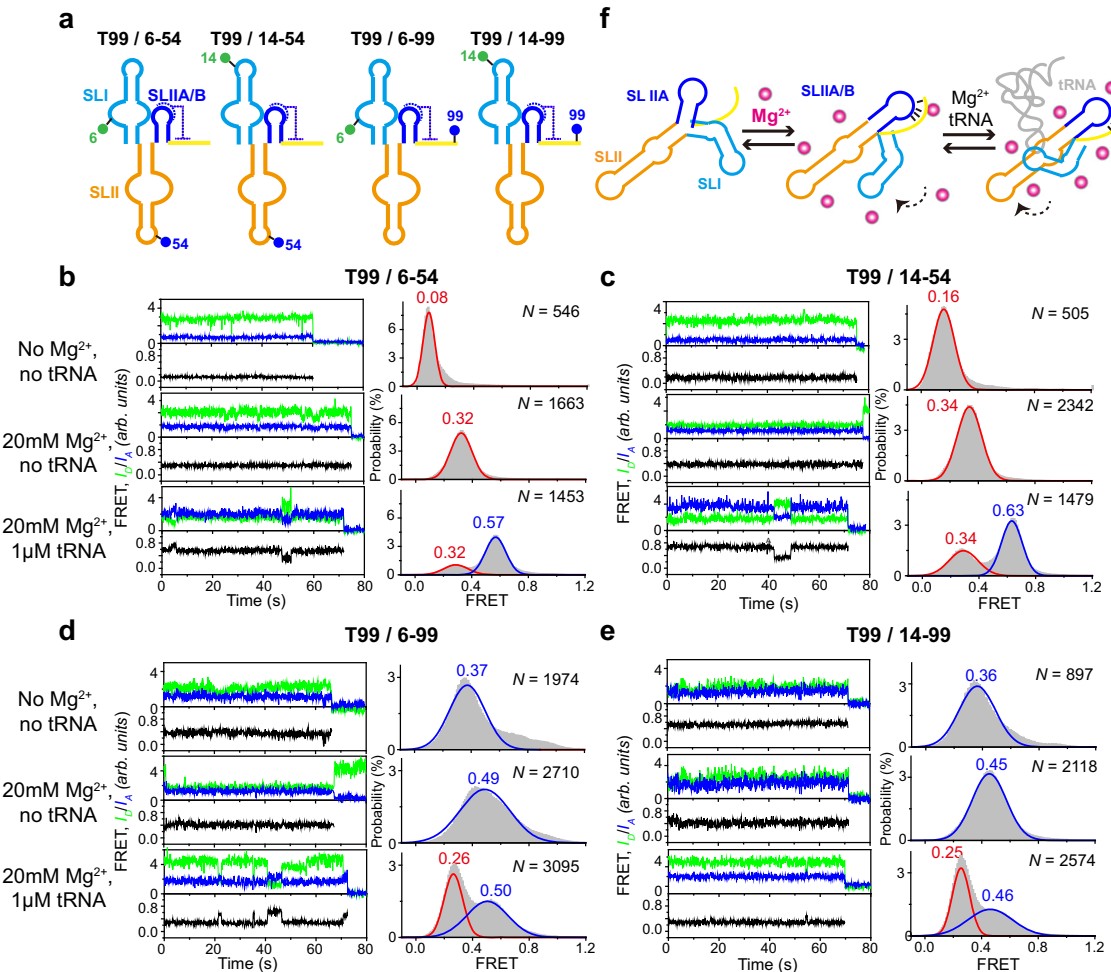

**Fig. 4 | $Mg^{2+}$ and tRNA mediated inter-domain structural changes of T99 analyzed by smFRET. a** Schematic representation for the secondary structure of each smFRET construct. The blue and green circle indicates Cy5 and Cy3 fluorophore, respectively. **b**–**e** Representative smFRET traces (left) and FRET histograms (right) for each smFRET construct in **a** in three different conditions: No $Mg^{2+}$, 20 mM $Mg^{2+}$ and 20 mM $Mg^{2+}$ in the presence of 1 µM tRNA. Green, blue and black lines represent the Cy3 intensity, Cy5 intensity and FRET efficiency, respectively. Histograms were well fitted with the Gaussian peaks, shown in red and blue for the low- and middle-FRET states, respectively. $N$ denotes the total number of traces to generate histograms from three independent experiments ($n = 3$). **f** Cartoon showing the global structural rearrangement of T99 induced by $Mg^{2+}$ and tRNA. Source data for panels **b**–**e** are provided as a Source Data file.

than that of T99/6-54, indicating that these mutants sample a more heterogeneous conformation and destabilization of the pseudoknot impairs $Mg^{2+}$-induced pre-docking of stems I and II. Strikingly, both of the deletion mutants (T89/6-54 and T77/6-54) alone in 7.5 mM $Mg^{2+}$ present a major low-FRET population with the same peak value (~ 0.08) as that of T99/6-54 in the absence of $Mg^{2+}$, implying that stems I and II are undocked and locate far away from each other, thus complete disruption of the pseudoknot abolishes $Mg^{2+}$-induced pre-docking of stems I and II.

The docking dynamics of stems I and II of such mutants in 7.5 mM $Mg^{2+}$ and tRNA were also investigated using smFRET (Fig. 5c). Under such condition, T99/6-54 binds tRNA with high affinity and transits frequently between a low- and a middle-FRET state corresponding to the pre-docked and docked state, respectively. Similarly, all the single-point and deletion mutants also sample the middle-FRET state, though the occupation is significantly lower than that for T99. However, only T89/6-54 samples a middle-FRET state with the same peak value (~0.59) as that of T99/6-54, and the peak values of the middle-FRET state for all the other mutants are higher, probably due to the breathing effect. These results demonstrate the importance of the structural integrity of stem IIA/B in promoting the pre-docking of stems I and II for high-affinity tRNA binding. It's interesting to note that the population of the middle-FRET state for T89 is larger than that for T77, suggesting that

the presence of stem IIA in T89 slightly contributes to the docking of stems I and II, though weakly than that in T99.

## The S-turn is critical to the tRNA-induced docking of stems I and II

Previous structure analysis shows that A69-G70-A71 trinucleotide in stem II S-turn is expected to use these backbone interactions to guide the specifier into a helical, stacked conformation poised to recognize the tRNA anticodon[19]. To understand how and when these intramolecular RNA-RNA interactions occur and their roles in T-box folding and tRNA binding, three single-point mutants (A69U, G70U and A71U) that disrupt these interactions were constructed. SAXS data indicated that these single-point S-turn motif mutations cause little effects on the global folding of T99 RNA (Supplementary Fig. 5a−c).

The docking dynamics of stems I and II in such mutants were characterized by smFRET (Supplementary Fig. 5d, e). It's interesting to note that all the mutants alone sample the same low-FRET state as that of T99/6-54 in 7.5 or 20 mM $Mg^{2+}$, indicating that the mutations cause little effects on $Mg^{2+}$-induced pre-docking of stems I and II and consistently, the Specifier-S-turn binding cleft is not formed prior to tRNA binding. However, none of these mutants sample the middle-FRET state corresponding to the tRNA-bound conformation upon the addition of tRNA in 7.5 mM $Mg^{2+}$. Only G70U/6-54 samples a

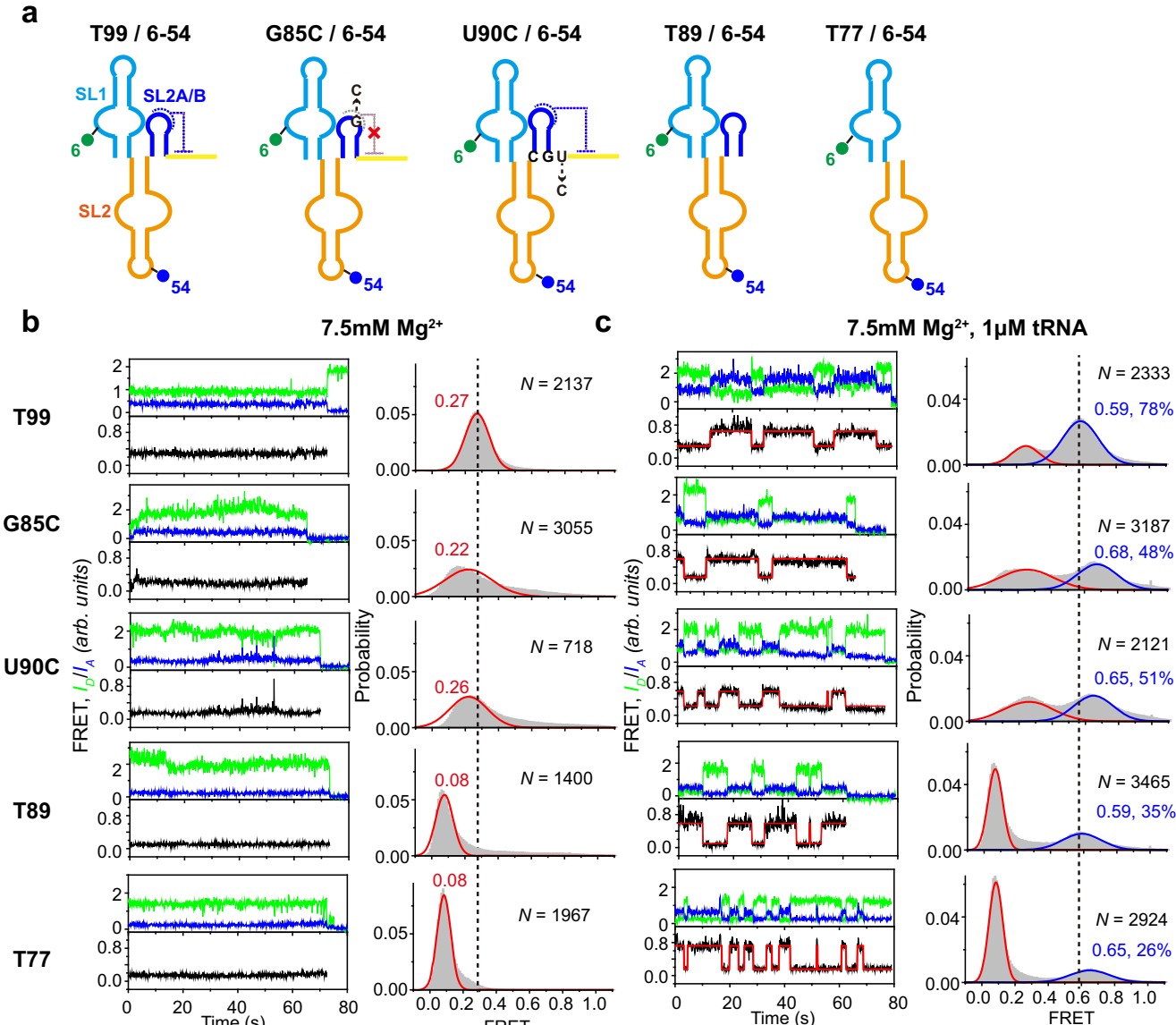

**Fig. 5 | Dynamic coupling between stem IIA/B and stem I. a** Schematic representation for the secondary structure of each smFRET construct. Sites for mutations to disrupt tertiary interactions are indicated. The blue and green circle indicates Cy5 and Cy3 fluorophore, respectively. **b–c** Representative smFRET traces (left) and FRET histograms (right) for each smFRET construct in **a** in the absence (**b**) and presence of 1 μM tRNA (**c**) in 7.5 mM Mg$^{2+}$. Green, blue and black lines represent the Cy3 intensity, Cy5 intensity and FRET efficiency, respectively. The Guassion fitting curves are shown in red above the experimental trace for T99 and mutants in the presence of tRNA. Histograms were well fitted with the Gaussian peaks, shown in red and blue for the low- and middle-FRET states, respectively. *N* denotes the total number of traces to generate histograms from three independent experiments (*n* = 3). Source data for panels **b–c** are provided as a Source Data file.

pronounced middle-FRET state (27%) in 20 mM Mg$^{2+}$, suggesting that high Mg$^{2+}$ attenuates the mutational effects of G70U. It's likely that the initial contact between the tRNA anticodon and Specifier in stem I drives the formation of a helical, stacked conformation for the specifier and then induces the docking of stem I towards stem II via backbone interactions with the S-turn region, which in turn reinforces the Specifier-anticodon interactions. Thus, the S-turn region is critical to stabilize the tRNA-induced docking of stems I and II.

## Mg$^{2+}$-dependence of the tRNA-induced conformational changes of T99

To better understand how Mg$^{2+}$-induced T99 folding and high-affinity tRNA binding are dynamically coupled, smFRET experiments were performed across a range of Mg$^{2+}$ concentrations for T99/6-54 (1–20 mM), T89/6-54 and T77/6-54 (2–50 mM) in the absence or presence of tRNA (Fig. 6). Individual smFRET traces were analyzed by two-state

Hidden Markov modeling (HMM) to quantitatively assess the structural and kinetic features of undocked and docked states of stems I and II.

Consistently, T99/6-54 in the presence of tRNA samples a low-FRET state that is observed in *apo*-T99/6-54, and an additional middle-FRET state across the full range of Mg$^{2+}$ concentrations (Fig. 6b), which presumably correspond to the unbound and tRNA-bound favored conformations, respectively. At low Mg$^{2+}$ concentrations, T99/6-54 mostly stays in the low-FRET state and rarely transits to the middle-FRET state. As Mg$^{2+}$ concentration increases, T99/6-54 samples the middle-FRET state more frequently for longer time periods (Fig. 6b, c). At above 4 mM Mg$^{2+}$, the middle-FRET state becomes predominant, indicating that T99 binds to tRNA efficiently. The FRET efficiencies of the low-FRET state increase from 0.19 to 0.34 monotonically from 1 to 20 mM Mg$^{2+}$, indicating that Mg$^{2+}$ promotes the pre-docking between stems I and II. By contrast, the FRET efficiencies of the middle-FRET

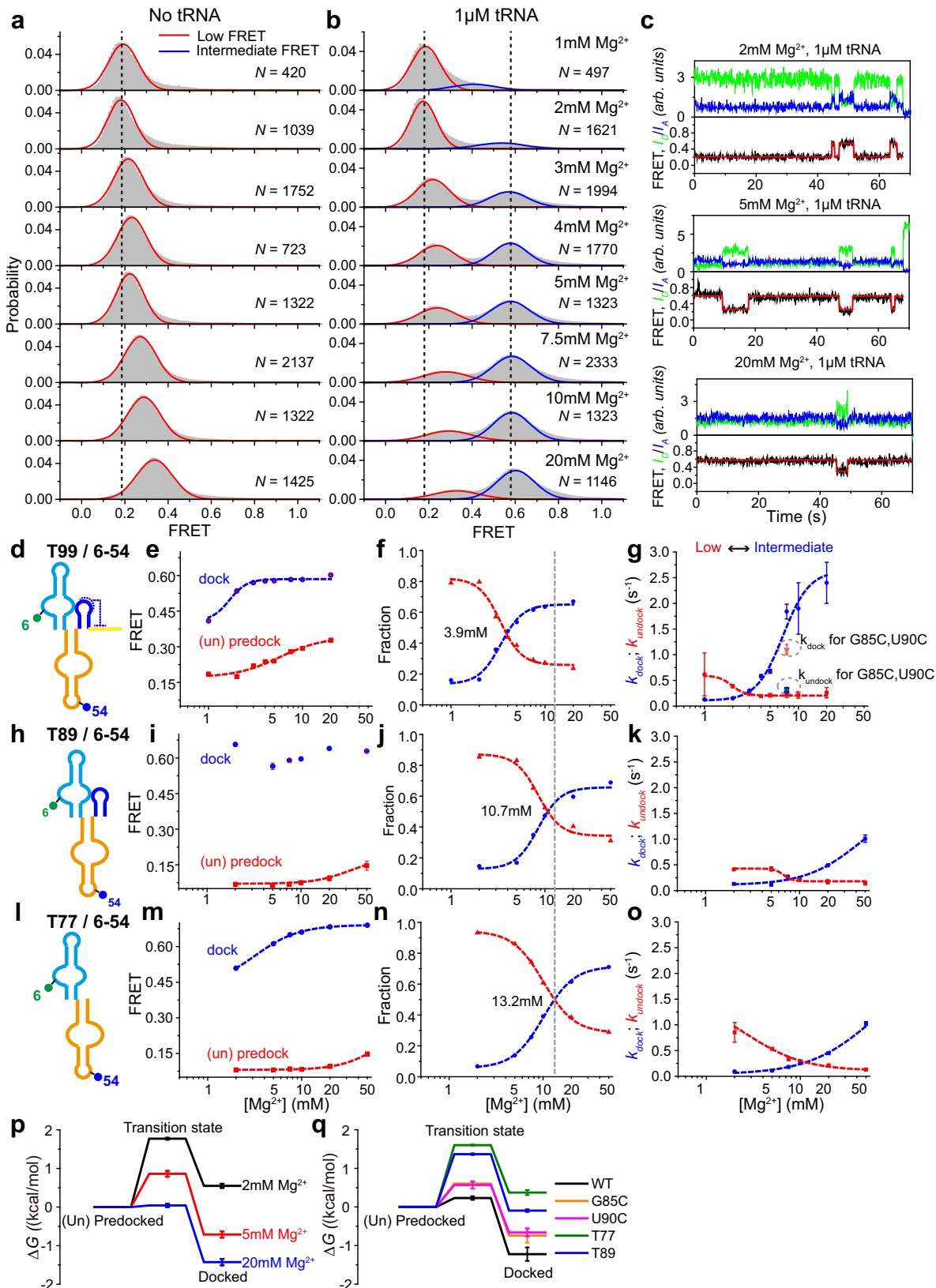

state increase obviously below 2 mM Mg²⁺ but then keep constant at higher Mg²⁺ (Fig. 6e), suggesting that the docked state is mildly affected by Mg²⁺. The total occupancy of each FRET state for T99/6-54 in the presence of tRNA is also calculated and plotted against Mg²⁺. The decrease of low-FRET state fraction was accompanied by an increase of middle-FRET state fraction as Mg²⁺ concentration increased. The Mg²⁺

concentration required for half-maximal binding ([Mg²⁺]₁/₂) is determined as $3.9 \pm 0.2$ mM using the Hill-equation fit, indicating that near-physiological concentration of Mg²⁺ is sufficient for high-affinity binding between tRNA and T99 (Fig. 6f). To analyze the docking and undocking kinetics, the dwell time distributions and transition rates between the low- and middle-FRET states were extracted from the

**Fig. 6 | Mg$^{2+}$ promotes the binding between tRNA anticodon and T-box riboswitch. a–b** FRET histograms for T99/6-54 in the absence (**a**) and presence (**b**) of tRNA across various Mg$^{2+}$ concentrations ranging from 1 mM to 20 mM. Histograms were well fitted with Gaussian peaks, shown in red and blue for the low- and middle-FRET states, respectively. $N$ denotes the total number of traces to generate histograms from three independent experiments ($n = 3$). **c** Representative smFRET traces for T99/6-54 in the presence of 1 μM tRNA in different Mg$^{2+}$ concentrations. Green, blue and black lines represent the Cy3 intensity, Cy5 intensity and FRET efficiency, respectively. The Gaussian fitting curves are shown in red above the experimental trace. **d–g** Schematic representation for the secondary structure (**d**), the centers of the Gaussian fits (**e**) and the fractional population of low- and middle-FRET state (**f**), and transition rate constants between low- and middle-FRET states (**g**) were plotted as a function of Mg$^{2+}$ for T99/6-54. **h–k** Schematic representation of the secondary structure (**h**), the centers of the Gaussian fits (**i**) and the fractional population of low- and middle-FRET state (**j**) and transition rate constants between low- and middle-FRET states (**k**) were plotted as a function of Mg$^{2+}$ for T89/6-54. **l–o** Schematic representation for the secondary structure (**l**), the centers of the Gaussian fits (**m**) and the fractional population of low- and middle-FRET state (**n**) and transition rate constants between low- and middle-FRET states (**o**) were plotted as a function of Mg$^{2+}$ for T77/6-54. The blue and green circle in **d**, **h**, **l** indicates Cy5 and Cy3 fluorophore, respectively. **p** Free-energy diagrams for tRNA-induced folding of T99 at various Mg$^{2+}$ concentrations. **q** Free-energy diagrams for tRNA-induced docking of T99 and stem IIA/B mutants in 7.5 mM Mg$^{2+}$. Data are presented as mean values ± SEM in **e–j**, **i–k**, **m–q**. Error bars are SEM from three independent experiments ($n = 3$). Source data are provided as a Source Data file.

individual single-molecule trajectories (Fig. 6g). The transition rates of the low- to middle-FRET ($k_{dock}$) increase significantly as Mg$^{2+}$ increases. In contrast, the reverse transition rates ($k_{undock}$) are less sensitive to changes of Mg$^{2+}$ concentrations, especially above 2 mM Mg$^{2+}$, which maintains at about 0.19-0.26 s$^{-1}$. The transitions between low- and middle-FRET states are likely caused by the tRNA binding and dissociation events since such transitions were rare for *apo* T-box. The free energy landscapes of T99 at different Mg$^{2+}$ concentrations in the presence of tRNA were estimated from transition rates between the undocked and docked states, revealing that Mg$^{2+}$ stabilized the docked state and significantly reduced the energy barrier from undocked to docked states (Fig. 6p). These data suggest that Mg$^{2+}$ promotes the high-affinity binding of tRNA to T99 mainly through enhancing the stability of the docked state.

To further explore the tRNA recognition mechanism by T99, we also performed smFRET experiments for T99/6-54 with varying concentrations of tRNA (1 nM-2 μM). As tRNA concentration increases, the occupancy of low-FRET decreases and middle-FRET increases (Supplementary Fig. 6) and the estimated $K_d$ (~40 nM) was comparable to the affinity reported for unlabeled T99. In addition, the transition rate $k_{dock}$ increases significantly while $k_{undock}$ changes slightly when tRNA concentration increases from 10 nM to 2 μM, suggesting that tRNA recognition by T99 favors an induced-fit mechanism.

By contrast, both T89/6-54 and T77/6-54 alone mainly populated a low-FRET state across the full range of Mg$^{2+}$ concentrations (2-50 mM), of which the peak value (~ 0.08) is insensitive to the increasing Mg$^{2+}$ and the same as that of T99/6-54 in the absence of Mg$^{2+}$ (Fig. 6h–o, Supplementary Fig. 7). These data suggest that the stems I and II in both constructs adopt an undocked conformation which is less affected by Mg$^{2+}$. It's likely the impairment of stem IIA/B hampers Mg$^{2+}$-induced pre-docking of stems I and II observed in T99. In the presence of tRNA, a middle-FRET state emerged at high Mg$^{2+}$ and its population gradually increased with the increasing of Mg$^{2+}$. The [Mg$^{2+}$]$_{1/2}$ calculated for T89/6-54 and T77/6-54 are 10.7 ± 0.3 mM and 13.2 ± 0.3 mM, respectively, which are ~3 and 4 folds as much as that for T99. Presumably, the T77-tRNA or T89-tRNA complexes are less stable in physiological Mg$^{2+}$, but can be stabilized by the crowded cellular environment as well as high Mg$^{2+}$. Additionally, the FRET efficiency for the tRNA-induced docking state becomes larger (from 0.51 to 0.69 in T77) as Mg$^{2+}$ increases. These data suggest that high Mg$^{2+}$ attenuates the detrimental effects of stem IIA/B disruption and promotes the docking of stems I and II in both T89/6-54 and T77/6-54 in the presence of tRNA. Further transition kinetic analysis shows an obvious increase of the $k_{dock}$ and decrease of the $k_{undock}$ in response to increasing Mg$^{2+}$, suggesting that Mg$^{2+}$ not only promotes the formation of the docked state, but also increases its stability by reducing the undocking rate (Fig. 6o).

As shown in Fig. 6g, the $k_{dock}$ decreased dramatically while the $k_{undock}$ increased mildly in the mutants (G85C, U90C) compared with T99. Thus, we speculated that the destabilization or deletion of stem IIA/B mainly increases the accessibility of stem I in the conformational space of T99 and finally results in their lower docking rate with tRNA.

This is also consistent with previous ITC data on the stem IIA/B mutants, in which they exhibited a larger entropic loss while the enthalpy change remained almost the same compared with T99[19]. Free energy landscapes for the stem IIA/B mutants and the T99 in 7.5 mM Mg$^{2+}$ were calculated, which showed that destabilization or removal of stem IIA/B significantly reduced the stability of the docked state, accompanied with an increase of the transition energy barrier from undocked to docked state by about 0.33-1.37 kcal/mol (Fig. 6q).

### 3D structural visualization of T99 transcriptional intermediates

The RNA constructs of T77 and T89 also represent the transcriptional intermediates of T99, thus can be used to investigate which structural features form as the RNA is being transcribed and extruded through an exit channel in the RNA polymerase. Similar to T99, the global folding of T89 is Mg$^{2+}$-dependent, but Mg$^{2+}$ has a minimal effect on the folding of T77, as evidenced by the small changes of $R_g$ and $D_{max}$ (Supplementary Fig. 1). To have a direct structural visualization of these RNA constructs, possible conformations that are consistent with their scattering profiles were computationally modeled by following the procedures for MD simulations in the Supplementary Methods. Individual structures that fit the SAXS data were selected and the theoretical scattering curves for all of the best-fit models are consistent with the respective experimental scattering curves (Supplementary Fig. 8, Supplementary Data 2). For T77, the best-fit models at low and high Mg$^{2+}$ are similar, of which the stems I and II are coaxially stacked (Supplementary Fig. 9) and predicted to give a low-FRET signal in T77/6-54 alone. For T89, the stem IIA hairpin is projected between stems I and II at low Mg$^{2+}$, probably due to the electrostatic repulsion, thus exhibiting a relatively extended conformation. At high Mg$^{2+}$, the negative charges on the RNA backbone can be effectively shielded, thus the stem IIA is more inclined to stack with stem II and slightly drives the reorientation of stem I toward stem II, resulting in a more compact conformation. For the best-fit model of T99 at low Mg$^{2+}$, the stems I and II are coaxially stacked but stem IIA/B pseudoknot is not formed, thus the stem IIA and 3'-end tail are highly dynamic, and both T99/6-54 and T99/14-54 are predicted to produce a low-FRET efficiency. By contrast, stem IIA/B pseudoknot at high Mg$^{2+}$ is predominantly formed and stacked against stem II (Supplementary Fig. 9), which further drives the pre-docking of stem I towards stem II, thus producing a low-to-middle FRET efficiency in T99/6-54, which agree with our smFRET measurements.

## Discussion

The T-box riboswitches are unique riboregulators in that gene regulation is mediated through interactions between two highly structured RNAs. A deep understanding of the folding pathway and conformational dynamics of the T-box riboswitches is of essential importance for not only deciphering the mechanisms and functions of RNA-RNA interactions but the development of novel antimicrobial therapeutics. In the present work, we have investigated in-detail the conformational dynamics of the aptamer domain and its mutants of a translational T-box riboswitch in response to Mg$^{2+}$ and tRNA and

delineated its folding pathway to achieve functional conformations by integrating ITC, SAXS, smFRET and computational modeling.

Our results demonstrate that $Mg^{2+}$-induced folding and high-affinity tRNA binding of the T-box aptamer are dynamically coupled. To achieve functional conformations, the nascent RNA generally need to proceed through a folding pathway, which could initially yield a large pool of partially folded conformations and nonfunctional states[33]. We find that physiological concentrations of $Mg^{2+}$ govern the folding pathway of T99 towards the tRNA binding competent conformations. Our SAXS and ITC data show that T99 adopts an unfolded conformation and is unable to bind with tRNA in the absence of $Mg^{2+}$, but the global shape of T99 becomes more compact and tRNA binding affinities become stronger as the $Mg^{2+}$ concentrations increase (Fig. 1). It's worthy mentioning that the large enthalpy change in 2 mM $Mg^{2+}$ is likely related to $Mg^{2+}$-induced preorganization of T99 (Supplmentary Table 1), which is a prerequisite for subsequent tRNA binding. However, even in the presence of high $Mg^{2+}$, T99 alone remains less compact than its *holo*-form, indicating that tRNA binding causes further conformational compaction in T99. Such coupled $Mg^{2+}$-induced folding and ligand binding mechanisms have been observed for several other riboswitches[34–36].

To enable 3D mapping of the conformational dynamics of the T99 by smFRET, we develop a site-specific dual fluorophore labeling scheme by taking advantages of the newly synthesized $rNaM^{CO}$ and the previously reported $TPT3^A$, which overcomes the limitations on RNA

size and labeling sites, as well as on the low labeling efficiencies of traditional labeling methods such as chemical synthesis and splint ligation. Empowered by the UBP-based scheme, comprehensive fluorophore labeling of T99 can be easily achieved with high labeling efficiency, allowing for in-detail mapping of the global and local structural dynamics and transition kinetics of T99 and its mutants in response to $Mg^{2+}$ and tRNA binding. While sophisticated matrix of distance network has been widely applied to study the conformational movement of membrane proteins by the double electron–electron resonance spectroscopy[37], such schemes have been less pursued in smFRET measurements of RNA, probably due to the lack of an efficient and universal RNA labeling method. We expect that the UBP-based labeling strategy will have broader applications in smFRET-based conformational dynamics visualization and structural modeling of large RNAs in solution.

RNAs tend to fold immediately after emerging from RNA polymerase during transcription. Cotranscriptional folding of riboswitches is highly related with ligand binding and time-dependent regulatory functions[38]. Our study on the intermediate transcripts of T99 suggested that tRNA decoding by *ileS* T-box riboswitch likely occurs in a cotranscriptional manner (Fig. 7). Due to the linear configuration of the three helices in T99, the stem I is transcribed first before the synthesis of stem II and stem IIA/B and may transiently interact with tRNA anticodon though the binding affinity is weak[19]. As stem II is transcribed from RNAP, it tends to coaxially stack with stem I and guide the folding of

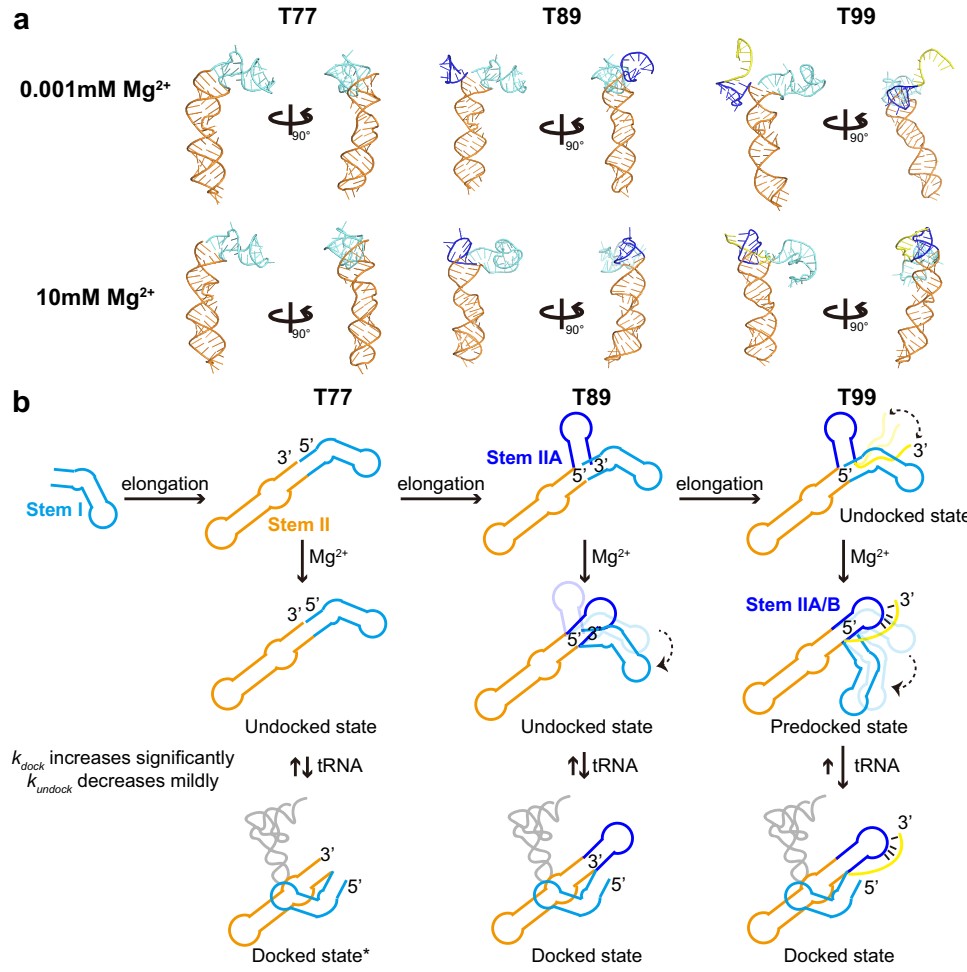

**Fig. 7 | Proposed model for $Mg^{2+}$ mediated T-box riboswitch folding and tRNA anticodon recognition. a** Atomic structural models for T77, T89 and T99 in low and high $Mg^{2+}$ which were derived from modeling and best-fitted with the respective SAXS scattering curves were shown in two different views for better visualization of the T-box structural rearrangement. **b** Proposed model for $Mg^{2+}$ mediated T-box riboswitch folding and tRNA anticodon recognition in a cotranscriptional manner.

T-box into an extended conformation as inferred by SAXS and smFRET. At that time, tRNA binds to T-box and slowly induces the docking of stems I and II. Further transcription of stem IIA guides the orientations of stem I towards stem II in physiological Mg²⁺. Both T77 and T89 require to overcome larger energy barriers for the transitions from undocked to predocked states. Only when the 3'tail is transcribed and forms compact pseudoknot with stem IIA, then the T-box riboswitch will bind tRNA anticodon efficiently and transit from the predocked to docked conformation quickly. In this way, we conclude that stem IIA/B is a hub structural element that could delicately preorganize the folding of T99 during transcription for efficient tRNA decoding. However, the tRNA-binding induced docking behavior in *ileS* T-box even at short transcript lengths, is different from that in *glyQS* T-box riboswitch, in which the tRNA capture and decoding is accomplished by a long stem I and tRNA decoding likely begins only when stem I is almost fully transcribed. The early binding of ligand by fluoride riboswitch during transcription has been found to impact the downstream gene expression[39]. These results provide important insights into the co-transcriptional folding pathways for T99. However, it's unclear whether the tRNA binding by *ileS* T-box at the early stage of transcription is related to the regulatory function and awaits further co-transcriptional investigation on the full-length T-box.

Our smFRET-based analysis of the intra- and inter-stem dynamics of T99 unveils the molecular mechanism to sculpt its high-affinity tRNA binding site. Riboswitches might recognize their cognate ligands through lock and key, induced-fit or conformational selection mechanisms, which can be regulated by Mg²⁺ due to its role in promoting RNA folding[36]. Previous structural analysis suggested a pre-formed tRNA anticodon binding groove created by the perpendicularly oriented stems I and II[19]. Different from this "lock and key" docking model, our results suggest an induced-fit recognition mechanism that Mg²⁺-induced pre-docking of stems I and II is a prerequisite for efficient initial tRNA contact to the specifier, but not sufficient to induce the formation of backbone interactions between the specifier and S-turn regions, since T99/6-54 at high Mg²⁺ mainly gives a low-to-middle-FRET state and rarely populates the middle-FRET. Subsequent tRNA binding induces further docking of stems I and II, resulting in a middle-FRET state in T99/6-54 consistent with the tRNA-bound crystal structure. While our manuscript was in revision, *Suddala* et al. reported a related single-molecule study on tRNA binding to the *ileS* T-box aptamer and proposed a similar "Venus flytrap" mechanism[40]. They observed an open conformation, a semi-open conformation and a minor closed conformations (10%) for *apo* T-box. tRNA mostly binds the open and the semi-open conformations to induce folding and formation of tRNA-bound state, which also suggest a predominant induced-fit mechanism. This differs from the Mg²⁺-promoted conformational selection mechanism reported in small-molecule sensing riboswitches such as preQ₁ and fluoride riboswitch[36,39].

Our work highlights a delicate balance among Mg²⁺, the intra- and intermolecular RNA-RNA interactions in modulating the cooperativity in T99 to regulate its folding and function. Mg²⁺ is essential for neutralizing the negative charge of phosphate moieties in the RNA backbone, which induces structural changes that bring together two RNA regions[32,41], thus allowing for long-range tertiary interactions and quaternary interactions to form[42–44]. Our study suggests strong cooperativity among the tertiary interaction networks involved in stem I Specifier region, stem II S-turn region, stem IIA/B pseudoknot and Mg²⁺ binding in T99, in which stem IIA/B plays an essential role. At low Mg²⁺, the stem IIA/B pseudoknot is not formed, and stems I and II are coaxially stacked, preventing the pre-docking of stems I and II to form a competent conformation for efficient tRNA binding. Increasing Mg²⁺ stabilizes the stem IIA/B pseudoknot formation, which drives the coaxial stacking of stem IIA/B on stem II, resulting in pre-docking of stem I on II into a competent conformation for tRNA binding. Subsequent tRNA anticodon binding to the specifier region induces further docking of stem I with stem II S-turn region. The cruciality of stem IIA/B and stem II S-turn in the pre-docking and

docking of stems I and II for high-affinity tRNA binding, respectively, was substantiated by mutational analysis. However, high Mg²⁺ can compensate for the destabilizing effects of stem IIA/B and stem II S-turn mutations which are destructive to the docking of stems I and II at low Mg²⁺. The intricate interplay among these interactions is directly visualized by computational modeling of the transcriptional intermediates of T99, showing how different structural modules cooperate with each other to promote RNA folding into functional conformations for efficient and fast tRNA recognition (Fig. 7).

Due to their roles in the regulation of multiple essential genes, T-box riboswitches are attractive drug targets. Recent advances in structural and biochemical studies have promoted the identification of several small molecules such as targeting the tRNA binding regions to disrupt gene regulation[45]. While structure-based virtual screening offers a powerful tool to screen potential ligands, however, only high-resolution static structures with a limited number of states are available, which often fails to identify reliable druggable pockets. Since the T99 aptamer and many other riboregulators exist as conformational ensembles containing dynamically interconverting substates, accurate characterization of the conformational ensembles under diverse conditions by combining smFRET, SAXS and computational modeling and targeting the RNA dynamic ensembles represents a potential avenue for the discovery of novel antibiotic[46–48].

## Methods

### RNA sample preparation
The wild type T99 and its mutant constructs were generated as follows. Plasmids coding an upstream T7 promoter and T99 were gene synthesized and sequenced by Wuxi Qinglan Biotechnology Inc, Wuxi, China. All the other mutants were generated using the Transgen's Fast Mutagenesis System. The double-stranded DNA fragment templates for in vitro RNA production were generated by PCR using an upstream forward primer targeting the plasmids and a downstream reverse primer specific to respective cDNAs. The RNAs were transcribed in vitro using T7 RNA polymerase and purified by preparative, non-denaturing polyacrylamide gel electrophoresis, the target RNA bands were cut and passively eluted from gel slices into buffer containing 0.3 M sodium acetate, 1 mM EDTA, pH 5.2 overnight at 4 °C. The RNAs were further passed through the size exclusion chromatography column to the final buffer condition for SAXS experiments. The sequences for all the constructs and the mutants are listed in Supplementary Table 2. The primer sequences used in this study are summarized in Supplementary Table 3.

### Small-angle X-ray scattering (SAXS) experiment
All the experimental procedures and instrument parameters for data collection and software employed for data analysis are similar as described before[49]. Briefly, SAXS measurements were carried out at room temperature at the beamline 12 ID-B of the Advanced Photon Source, Argonne National Laboratory. The scattered X-ray photons were recorded with a PILATUS 2 M detector (Dectris) at 12 ID-B. The setups were adjusted to achieve scattering $q$ values of $0.005 < q < 0.89$ Å⁻¹, where $q = (4\pi/\lambda) \cdot \sin(\theta)$, and $2\theta$ is the scattering angle. Thirty-two-dimensional images were recorded and reduced for each buffer or sample and no radiation damage was observed. Scattering profiles of the RNAs were calculated by subtracting the background buffer contribution from the sample-buffer profile using the program PRIMUS3.2 following standard procedures[50]. Guinier analysis was performed to calculate the forward scattering intensity $I(0)$ and the radius of gyration ($R_g$), which were also estimated from the scattering profile with a broader $q$ range of 0.006–0.30 Å⁻¹ using the indirect Fourier transform method implemented in the program GNOM4.6[51], along with the pair distance distribution function (PDDF), $p(r)$, and the maximum dimension of the RNA, $D_{max}$. Low-resolution 3D shape envelopes were ab initio reconstructed using the scattering data within the $q$ range of

0.006–0.30 Å⁻¹ with the program DAMMIN[52], which generate models represented by an ensemble of densely packed beads. The theoretical scattering intensity of the atomic structure model was calculated and fitted to the experimental scattering intensity using CRYSOL[53].

## Isothermal titration calorimetry (ITC)

RNA samples used for Isothermal Titration Calorimetry experiments were prepared as described above and stored in a buffer containing 20 mM Tris (pH 7.4), 100 mM KCl supplemented with different concentrations of $Mg^{2+}$. ITC measurement was performed in triplicates at 25 °C, with ~20 μM T99 in the cell and ~200 μM tRNA in the syringe, using a MicroCal PEAQ-ITC microcalorimeter. The raw ITC data were analyzed using the Origin7 software package provided by the manufacturer to obtain the dissociation constant and thermodynamic parameters summarized in supplementary Supplementary Table 1.

## Electrophoretic mobility shift assay (EMSA)

RNA samples used for EMSA were prepared as described above and stored in a buffer containing 50 mM HEPES (pH 7.5), 100 mM KCl. tRNA (2.4 μM) was incubated with varying concentrations of dye-labeled or unlabeled T-box (0.24–7.2 μM) supplemented with 20 mM $Mg^{2+}$ in 37 °C for 30 min. Then the mixtures were loaded on 8% native polyacrylamide gel containing 10 mM $Mg^{2+}$ and the gel was run at room temperature for ~60 min. The gel was then stained by Gelsafe dye and imaged using gel imager. EMSA for different labeling constructs were conducted on independent biological triplicates.

## Site-specific internal labeling of RNAs using UBP system

To develop a site-specific dual fluorophore labeling scheme for large RNAs using the NaM-TPT3 unnatural base pair (UBP) system, we synthesize an alkyne-modified rNaMTP derivative (rNaM^CO) by following the procedures in the Supplementary Methods. The deoxyribonucleotide phosphoramidites (dTPT3-CEP and dNaM-CEP, for DNA primer synthesis), the triphosphorylated deoxyribonucleotides (dTPT3TP and dNaMTP, for PCR) and ribonucleotides (rTPT3^ATP, for transcription), were custom synthesized as described[29]. Reversed primers containing unnatural nucleotides were synthesized and used for overlapping PCR to introduce the dNaMTP and dTPT3TP into the DNA template. Then the PCR products containing dNaM and dTPT3 at specific sites were used as the DNA templates for in vitro transcription and purified as described above. The purified T99 products modified with rNaM^CO and rTPT3^A at specific sites were precipitated with cold ethanol (2.5 volumes) in the presence of sodium acetate (0.3 M) at −80 °C for at least 0.5 h. After centrifuging at 4 °C, the ethanol was removed and the pellet was washed with ethanol (75%) three times, then the pellet was dried for 10 min. Finally, the product was resuspended in DEPC-treated water and subjected to Cy5 fluorescent labeling. RNAs (0.3 mM, 10.5 μL) were mixed with 33.5 μL 1.5 X Click chemistry buffer, 5 μL Sulfo-cyanine5 azide (20 mM in DMSO), 1 μL of sodium ascorbate (50 mM). The resulting mixture was incubated at room temperature overnight. Then the labeled T99 are precipitated by ethanol as described above and resuspended in 0.1 M NaHCO₃ (pH 8.0), then the Cy5-labeled T99s were mixed with 5 μL NHS-Cy3 (20 mM in DMSO) and incubated at room temperature for 4–6 h. Then Cy5-Cy3 double labeled T99s were precipitated by ethanol as described above and resuspended in buffer containing 50 mM HEPES (pH 7.4) and 100 mM KCl, stored at −80 °C for further smFRET experiments. The labeling efficiency was calculated by measuring the absorption of RNAs at 260 nm, 546 nm and 650 nm, respectively. The labeling efficiencies for Cy3 and Cy5 are ~70% and ~90%, respectively.

## 5′-end fluorophore labeling of RNAs

5′-end fluorophore labeling of RNAs was performed by following the protocols developed in previous studies with a minor improvement[23,54]. Briefly, purified RNA was precipitated by ethanol as described above and resuspended in 0.1 M MES (pH 6.0). This was followed by the addition of N-(3-Dimethylaminopropyl)-N0-ethylcarbodiimide hydrochloride (EDC) along with ethylene diamine and 0.1 M imidazole solution (pH 6.0). The mix was incubated at 37 °C for 3 h and followed by ethanol precipitation for at least three times to remove residual EDC. Then the EDC-treated T99 was resuspended in 0.1 M NaHCO₃ (pH 8.0) and mixed with 30 folds NHS-Cy3 dye, incubated at room temperature for 4–6 h. Then Cy5-Cy3 double labeled T99s were precipitated by ethanol as described above and resuspended in buffer containing 50 mM HEPES (pH 7.4) and 100 mM KCl, stored at -80 °C for further smFRET experiments.

## Single-molecule FRET experiments

For single-molecule experiments, 450 nM double labeled T99s were annealed with 300 nM biotin-modified DNA in 50 mM HEPES (pH 7.5), 100 mM KCl by incubating the mixture at 95 °C for 2 min, then fast cooling on the ice and add $Mg^{2+}$ to the final concentration of 20 mM, and finally equilibrated at 37 °C for 20 min. Samples were diluted 1200 times in the buffer containing 50 mM HEPES, 100 mM KCl with different concentrations of $Mg^{2+}$ and immobilized on the slides by biotin-streptavidin interactions. Then the samples were incubated with different concentrations of $Mg^{2+}$ or tRNA for 5 min on the slide before flowing the imaging buffer containing 3 mg/mL glucose, 100 μg/mL glucose oxidase, 40 μg/mL catalase, 1 mM cyclooctatetraene (COT), 1 mM 4-nitrobenzylalcohol (NBA) and 1.5 mM 6-hydroxy-2,5,7,8-tetramethyl-chromane-2-carboxylic acid (Trolox). smFRET experiments were conducted at 25 °C by using a home-built objective-type TIRF microscope. The time resolution of each movie was 100 ms/frame and we collected 800 frames for each movie. In each field of view, about 50-60% molecules were selected for further analysis. 3-4 movies were collected for each conditions. All of the experiments were repeated at least three times, from which experimental errors were estimated.

## Single-molecule FRET data analysis

smFRET data were analyzed by the custom-made software program. Single-molecule movies were collected by Cell Vision software (Beijing Coolight Technology) and then analyzed by a custom-made software program developed as an ImageJ 1.43 u plugin (http://rsb. info.nih.gov/ij). Fluorescence spots were fitted by a 2D Gaussian function within a 9-pixel by 9-pixel area, matching the donor and acceptor spots using a variant of the Hough transform[55]. The background-subtracted total volume of the 2D Gaussian peak was used as raw fluorescence intensity $I$. FRET trajectories containing donor and acceptor and displayed anticorrelation behaviors were picked and analyzed. Histograms of FRET efficiency for T-boxes at different concentrations of $Mg^{2+}$ or tRNAs were built by using the frames of all traces before photobleaching (more than 400 molecules, on average >250 frames per molecule). The histograms were divided by the total number of FRET frames and the frequencies were labeled as 'probability' in the $y$-axis.

smFRET traces were further analyzed by the Hidden Markov Model (HMM)-based software to extract the kinetics information[57]. Two FRET states from low- to high- FRET values were identified as low- (undocked and docked state) and middle-FRET (docked state). FRET populations for two states derived from HMM analysis were fitted to Gaussian using Origin, and the relative fraction of each FRET state was calculated as the ratio of each FRET state to the total population. The dwell times of different FRET states were calculated from idealized traces and the dwell time distribution was fitted to a two-exponential function to calculate the transition rates. Relative free energies were calculated through the equation $\Delta G_b - \Delta G_a = -k_B T \bullet \ln\left(\frac{k_{a\rightarrow b}}{k_{b\rightarrow a}}\right)$, $k_B$ is the Boltzmann constant, $T$ is the temperature, $k_{a\rightarrow b}$ and $k_{b\rightarrow a}$ are the transition rates from the state a to b and from the state b to a, respectively. L state was set as the ground state ($\Delta G_{Low} = 0$). For illustration, the energy barrier from the state a to b was calculated as $-k_B T \cdot \ln(k_{a\rightarrow b}) + 1.8 k_B T$.

## Computational modeling

We exploited all-atom explicit solvent MD simulations to generate a reasonable conformational pool for post-simulation screening the model satisfying SAXS data on T99, T89 and T77 RNA. The details regarding the preparation of initial structural models for MD simulation (Supplementary Data 1) and MD simulation control parameters can be found in Supplementary Methods. After the MD simulation, the program Crysol was utilized to calculate the scattering intensity profile for all structure models from MD simulations and fit the back-calculated scattering intensity profile to the experimental observation. The $q$ range 0.006–0.3 Å$^{-1}$ with a pace of 0.002 Å$^{-1}$ was used in our all calculations and fittings, and the other parameters were default. The best-matching model (with a smaller $\chi^2$) was selected.

## Reporting summary

Further information on research design is available in the Nature Portfolio Reporting Summary linked to this article.

## Data availability

The data that support the findings of this study are provided in the Supplementary information, Supplementary Data files and Supplementary Movies including both the experimental data and the computational data. All other data are available from the corresponding author upon request. Source data are provided with this paper.

## Code availability

Custom scripts used to analyze the single-molecule FRET data are available upon request.

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

## Acknowledgements

We thank the staffs at beamline 12-ID-B, Advanced Photon Source, Argonne National Laboratory during SAXS data collection. This work was supported by grants from the Strategic Priority Research Program of the Chinese Academy of Science, Grant No. XDB0570000 to X.F., the National Natural Science Foundation of China (No. U1832215 to X.F., No. 21922704, 22277063 and 22061160466 to C.C.), the National Key Research and Development Project of China (2021YFA1301500 to X.F.), the Beijing Frontier Research Center for Biological Structure to X.F. and C.C., China Postdoctoral Science Foundation (No. 2022M711845) and Shuimu Tsinghua Scholar Program to X.N.

## Author contributions

X.F. conceived and designed the project. C.C. and X.F. supervised the single-molecule FRET experiments. X.N. performed the single-molecule FRET experiments and analyzed the data. X.N. prepared SAXS sample and analyzed the SAXS data. X.Z. performed SAXS experiments. Z.X. performed the MD simulations. Y.Z. performed the ITC experiments. X.N., C.C. and X.F. wrote the manuscript with inputs from the other authors.

## Competing interests

The authors declare no competing interests.
