## [Peer Review File · Nature Communications]

Structural and dynamic mechanisms for coupled folding and tRNA recognition of a translational T-box riboswitchREVIEWER COMMENTS

Reviewer #1 (Remarks to the Author):

Presented is a comprehensive study of the conformational properties of the translational T-box riboswitch and its interaction with tRNA. This is an interesting model system to study RNA as well as a potential target for the development of antimicrobial therapies. The paper is well written and the results presented in a clear and concise manner. The comprehensive site-specific labeling approach to map details of the overall conformational properties of the RNA using FRET is quite powerful, offering a very detailed picture of the different states sampled by the system and how Mg²⁺ impacts those states. Notably, the labelling approach will be useful to other labs working in this area, thereby more broadly facilitating studies of the conformational properties of RNAs. Thus, the manuscript is of broad interest in a number of ways, making it suitable for Nature Communications. A number of issues that should be addressed prior to publication follow.

- 1) Line 57: Use of the word "hysteretic" is not correct. Consider "limited"
- 2) The authors should reference some of the original articles by David Draper on the role of Mg²⁺ in driving formation of the tertiary structure of RNA. For example, this is consistent with the observation in the present study "that Mg²⁺ has a minimal impact on the folding of individual stems I and II, but is required for the formation of stem IIA/B."
- 3) It's not clear if the binding affinity to tRNA of all the modified

T-box RNAs used for the FRET analysis were tested and reported. Please include this data.

4) For easy reference it would be helpful to report the concentration of the RNA species used in the experiments be included in the figure legends.

5) Structures associated with Figures, especially Figure 7, need to be included in the supporting information to allow readers to analyze the structures in detail. Structures used to initiate the MD simulations should also be supplied to allow others to more readily reproduce the calculations.

6) Line 526: "incompact" to "less compact"

7) SI: The description of the computational methods for the RNA equilibration and production simulations needs to be significantly improved to allow other users to repeat the calculations. Details of the restraints used, atom truncation etc. are required.

Reviewer #2 (Remarks to the Author):

This very thorough study of the folding pathway of the T-box riboswitch shows presents a model of how Mg²⁺-dependent assembly of the aptamer domain of the riboswitch leads to binding of the tRNA anti-codon. The results also show that tRNA binding induces another change in the riboswitch conformation. Thus, as is common in regulatory RNA elements, ligand recognition occurs in multiple stages.

The strength of this study is the combination of SAXS, smFRET, and some MD modeling, which together provide a clear picture of how the RNA structure changes in response to Mg²⁺ and tRNA binding. Another outstanding aspect of this study is the use of non-standard bases to incorporate the desired fluorophores at different sites within the riboswitch. This strategy allowed the authors to test four different labeling schemes as well as several mutations, resulting in an unusually large amount of smFRET data. The data are high quality, and apart from some technical points noted below, the results are presented completely and clearly.

The weakness of the study design is that the authors only investigated folding of the aptamer domain, and not the full-length riboswitch. This is a shame because recognition of the tRNA acceptor is at the heart of the regulatory mechanism. In addition, the concentration of tRNA wasn't varied, making it more difficult to assess how the T-box responds to its ligand.

Overall, this study adds to the examples of riboswitch folding pathways. The details of the folding pathway of this translational T-box riboswitch are likely to be useful for those studying these riboswitches. The concepts of folding and riboswitch organization are established, however, so in that sense, this study doesn't break new ground.

Specific comments:

1. The ITC results on the tRNA binding in Fig. 1g are very unusual, and demand more explanation. The non-monotonic variation in ΔH for tRNA binding in different Mg²⁺ concentrations indicates a large change in the tRNA binding pathway at high and low Mg²⁺. This change could be related to preorganization of the aptamer as outlined in Fig. 7 but the authors don't comment on these data at all. The titration with magnesium is not fine enough to know whether the curve at 2 mM Mg²⁺ is a fluke, or a complex trend. Alternatively, the ITC results can be taken out of this manuscript, since they add very little.
2. Although the smFRET data in Fig. 3 and Fig. 4 appear high quality and are nicely presented, the unliganded T99 RNA does not show any transitions in the few example trajectories (except when tRNA is added). Yet, the authors describe the RNA structures as dynamic, and the population histograms in Fig. 3b (for example) show more than one FRET state. Do these low FRET states arise from heterogeneity among the RNA molecules? The authors should include more examples in the SI and describe the origins of these FRET peaks more clearly.

3. Related to the point above, the helices are cartooned as fluctuating between different orientations in Fig. 4f. I find this plausible. However, the basis for this conclusion is not well explained or justified. Do the authors assume that these fluctuations are faster than 0.1 s, and thus broaden the FRET distribution but are not resolved in the movies? Or are they very slow, and thus rarely happen to a given RNA within the span of 20-80 s? These alternatives make a difference to the model for tRNA recognition.

This point can be addressed by collecting a few longer movies. If photobleaching cannot be reduced, one can acquire data for a long period (5-10 min) by shutting off the excitation between recording intervals. Another way to address it is to jump the magnesium concentration between high and low values while recording, to see how rapidly the RNA population responds. If the response is immediate, it says that the conformational equilibrium is in fast exchange relative to the imaging rate. If some molecules respond and others don't, it suggests a fraction are trapped in a non-productive structure.

4. A description of how the RNA quality was validated is missing. What proportion of molecules contained both fluorophores? Does T7 RNAP sometimes mis-incorporate opposite the non-standard template base, or does it just stall? How was the RNA annealing protocol evaluated – for example, is the K_d for tRNA binding comparable to literature values? Can the T99 RNA be saturated with tRNA (>90%)? Does the full-length riboswitch prepared in this way respond to tRNA concentration in the expected range?

5. The Mg titrations in Fig. 6a,b are really beautiful and clearly show that tRNA binding induces a new conformational state of the riboswitch. On line 391, line 423 and in Figure 6, however, the authors mention that they have measured the tRNA binding kinetics. Do the authors assume that the high FRET state Fig. 6c represents tRNA binding? How do they distinguish binding from a conformational change in the tRNA-riboswitch complex? Please explain in the manuscript. Also, without varying the tRNA concentration, it is hard to estimate the on rate – this caveat should be also noted in the main text.

6. As I understand it, computed RNA models were selected that fit the SAXS data – was this done for an ensemble, or individual structures? Can the experimental curves be fit well by a single structure? Please explain this more clearly in the main text and methods.

7. Fig. 7 superimposes individual ribbons on the envelopes, which I presume come from bead models (DAMMIN). First, if more than one structure can fit the data, then this should be shown in Fig. 7 in some fashion. Second, the bead models are not the best way of interpreting the SAXS data, as the authors likely know. DAMMIN is a particularly poor choice for RNA that is more electron dense and that has different patterns of solvation and ion association than proteins. Did the authors try other software packages? Or, perhaps there is no need to show the bead models if they weren't used to evaluate the calculated RNA structures.

8. More description of the data analysis and its errors is needed – in addition to the number of molecules analyzed, the authors should also provide some estimate of the errors in the HMM fitting, the proportion of molecules in each FOV that were used. For the population histograms, it would be helpful to know how many frames of each movie were used.

9. The introduction is well written but quite long (as is the discussion). To help readers appreciate their work, the authors may want to consider saving some of this background material for a review article, and instead focus on the translational T-box riboswitch they have studied. It is not until line 116 that one learns that folding of a translational T-box (vs transcriptional T-box) is the new question here!

10. To aid understanding, please avoid acronyms, and if they must be used, be sure to define them in the main text. I found the acronym “RRI” particularly unhelpful and unnecessary.

11. A minor suggestion is to move the SAXS data in Figure 1 to Figure 7, where the data will be used to model the folding intermediates. Figure 1 is quite complicated, yet the results here were almost not used. The real focus of this paper is on the smFRET results and it would help to get to that immediately.

12. In the legend to Fig. 3 (and following), please explain that the blue lines represent Cy5 intensity and green lines represent Cy3 intensity.

13. The free energy diagrams in Fig. 6m,n show a main conclusion of the paper, yet this is almost buried in the amount of detail. I suggest moving the data for the mutants in Fig. g-l to the SI, so that more space and prominence can be given to the results that count.

Reviewer #3 (Remarks to the Author):

Review Comments for Structural and dynamic mechanisms for coupled folding and tRNA recognition of a translational T-box riboswitch by Niu et al.

The manuscript by Niu et al. utilizes a thorough combination of SAXS, smFRET, and molecular dynamics simulations to investigate tRNA decoding by the T-box riboregulator. T-boxes represent unique genetic regulatory elements that have analogous function to riboswitches but bind non-acylated tRNA instead of

small molecule ligands. By decoding and sensing the aminoacylation status of the bound tRNA, T-boxes facilitate regulatory responses to the depletion of specific amino acids. Here, the authors examine decoding with a minimal decoding module consisting of Stem-I, Stem-II, and Stem-IIA/B domains. A major strength of the study is the development and use of the unnatural base pair (UBP) system for multiple dual labeling schemes of the decoding module. This allowed them to survey using smFRET the relative motions of all domains within the module across Mg²⁺ concentrations and in the presence of tRNA. An important conclusion from this approach is that pseudoknot formation is critical for tRNA binding, presumably by sterically restricting the motion of Stem-I, thereby encouraging interaction between the specifier and Stem-II S-loop. Overall, the data within the manuscript are a very nice complement to recent high-resolution T-box structures and single-molecule experiments. However, in some cases the authors' conclusions could be presented more consistently and clearly. Questions and comments are below:

1. Something that bears mentioning in the manuscript is that in all experiments the decoding module is being studied at equilibrium. In the cell, these riboregulators function co-transcriptionally and likely sample conformations that cannot be captured in these experiments. For example, the tRNA will be able to interact with Stem-I specifier shortly after it is transcribed and before Stem-II or Stem-IIA/B are made. It would be very worthwhile for the authors to make mention of these caveats in the discussion and possibly frame some of their conclusions in light of how decoding may function co-transcriptionally. Framing the role of the Stem-IIA/B pseudoknot in a co-transcriptional scenario is especially important.

2. The prevailing model of riboswitch-ligand interaction is that the RNA can alternate between the ligand-bound and apo states even when the ligand is not present. Typically, these two states differ in their base-pairing scheme, and alternating between these two states tends to be rate-limiting, therefore the mechanism typically involves ligand-induced conformation capture of the correctly base-paired conformer. Here the authors propose an "induced-fit" model to define the action of the tRNA decoding module of T-box, where no alternative base-pairing is involved. The authors need to clearly define the importance of this induced-fit action in the context of the entire riboswitch to avoid confusing the readers that riboswitches may function entirely through an induced-fit mechanism.

3. The authors mainly use the secondary structure model to describe their induced-fit model. Given that high-resolution T-box/tRNA structures are available, it is more effective to use structural models to describe their mechanism. This reviewer encourages the authors to generate a morphing movie using Pymol or Chimera to describe their envisioned conformational changes. The authors could highlight the positions of the fluorophores on the model and their distances. Make sure to clearly define which state is hypothetical, which state is based on real structures.

4. Along the same line, is there any Mg²⁺ binding sites in the structure that could explain the higher FRET state in T-box in the absence of tRNA? For example, there is a mg bound by G41/G42 in the

published structure. Is this G-C pair conserved for the purpose of stabilizing the high FRET state? What happens when the G-C pairs here are changed to A-U pairs?

5. The effect of the magnesium concentration should be further discussed. Stabilization of the tRNA/T-box Stem-I specifier and Stem-II S-loop requires 10 mM Mg²⁺. However, the free [Mg²⁺] is typically reported in the 1-5 mM range, therefore the complex is expected to be less stable in vivo. The functional implications should be discussed.

6. The authors make two seemingly contradictory statements within the manuscript:

1. "Taken together, these smFRET data reveal a sequential docking mechanism for T99 that high Mg²⁺ facilitates the folding of stem IIA/B pseudoknot and its stacking on stem II for pre-docking of stems I and II to form a competent tRNA binding conformation, and subsequent tRNA binding drives further docking of stem I on stem II, and concomitantly, moving away from stem IIA/B."

2. "It's likely that the initial contact between the tRNA anticodon and Specifier in stem I drives the formation of a helical, stacked conformation for the specifier and then induces the docking of stem I towards stem II via backbone interactions with the S-turn region, which in turn reinforces the Specifier-anticodon interactions."

In statement 1, Stem-I and Stem-II preform a tRNA binding site while in statement 2, the tRNA binds Stem-I first before docking with Stem-II. The confusion between these two statements stems from the ambiguous definition of "pre-docking" in the manuscript. Does pre-docking refer to a preformed Specifier-S-turn binding cleft for tRNA? Does the low FRET state in high Mg²⁺ correspond to the pre-docked conformation? Why does this low FRET state still exist in the S-turn mutants? Clearing up these questions would greatly improve the clarity of the manuscript and models presented therein.

4. The figure of the 3D structure showing dye placements for the different constructs in the FRET experiments (Figure 2D) is difficult to interpret. A better representation would be to remove the dashed distance markers and simply label the different dye sites. The distances between the different dye pairs could be provided in a separate table.

5. The authors should consider changing "hysteretic" in the sentence "However, our understanding of how RRI occurs and drive the RNA folding and conformational dynamics remains relatively hysteretic." to something like "lacking" or "minimal".

Reviewer #4 (Remarks to the Author):

In the manuscript entitled “Structural and dynamic mechanisms for coupled folding and tRNA recognition of a translation T-box riboswitch” the authors present an analysis of the translational T-box regulator from *N. farcinica*. This is the first such study of which I am aware of a translational T-box, there are existing smFRET studies of the more prevalent transcriptional T-boxes (Suddala Nat. Com 2018, Zhang et al. eLife 2018). This work includes what appears to be some nice smFRET that is enabled by the incorporation of non-natural bases to allow the integration of dyes at specific sites. The method for incorporation looks to be quite flexible and is likely of use to many researchers, although it does look to have been previously published by same group (Wang et al. PNAS 2020). Not much time in the discussion is spent on the biological significance of the findings, and in particular how transcriptional and translational T-boxes may have similar distinct mechanisms for tRNA recognition. Overall, this is a nice study that adds knowledge to the growing cannon on T-box recognition, but ultimately does not fully contextualize its findings.

Major comments:

20 μM Mg^{+2} is high and not physiologically relevant. Physiological Mg^{+} is thought to be approximately 0.5 to 2 mM, and even the authors own data suggest that the Mg^{+2} concentrations in the range of 2-5 mM are sufficient for tRNA binding. It is not clear why the data in Figures 3 and 4 were collected at such a high concentration of Mg^{+2} . It may be that results 20 mM looks pretty much the same as lower concentrations (data on Fig. 4b (20 mM Mg^{+2}) and 5b (7.5 mM Mg^{+2}) look similar to each other), but it is clear that Mg^{+2} does have an impact on the folding as monitored by smFRET (Fig. 6ab), yet the authors capture only a non-physiological snapshot for much of the work conducted.

The authors have not substantiated this line from the discussion (pg 26 line 517). “To achieve functional conformations, the nascent RNA in general must proceed through a folding pathway, which could initially yield a large pool of partially folded conformations and nonfunctional states”. The authors assessed two truncations, and demonstrate that one of them (T77) is not competent to bind. This does not constitute characterization of the folding pathway for the RNA.

The biological impact of this work is missing from the text. What do we know now about mechanism, biology, or drugability, that we did not know previously? This work suggests that many of the findings are similar to those previously published (Suddala Nat. Com 2018, Zhang et al. eLife 2018), although there do seem some distinctions between this potentially concerted mechanism and the two-step mechanism described in prior works. The lack of contextualization decreases the general interest in the work substantially.

Minor comments:

The genus name of *N. farcinica* is never given in the main text.

The use of both T99 and WT is a bit confusing. I think these are the same construct, but both terms are used at different points.

A better indication on the cartoons in Figure 5 of the relationship between the fluorophores and the base changes would be nice. The U90C is indicated fairly clearly, but the G85C is not similarly indicated.

Figure 6 is difficult to interpret due to inadequate labelling with the figure and in accurate legend. Parts d-i correspond to different things in a 3x3 grid which is labelled d-f in the top row, g-i for the second row and j-l for the third row. However, the legend refers to "(d-j)", "(e-k)", and "(f-l)". These are not accurate. Also, as the illustration itself is not labelled itself, it is difficult to understand which constructs are represented in which figure part.

Response to reviewer 1

Referee #1:

Presented is a comprehensive study of the conformational properties of the translational T-box riboswitch and its interaction with tRNA. This is an interesting model system to study RNA as well as a potential target for the development of antimicrobial therapies. The paper is well written and the results presented in a clear and concise manner. The comprehensive site-specific labeling approach to map details of the overall conformational properties of the RNA using FRET is quite powerful, offering a very detailed picture of the different states sampled by the system and how Mg²⁺ impacts those states. Notably, the labelling approach will be useful to other labs working in this area, thereby more broadly facilitating studies of the conformational properties of RNAs. Thus, the manuscript is of broad interest in a number of ways, making it suitable for Nature Communications. A number of issues that should be addressed prior to publication follow.

Response #1: We thank the reviewer for the positive comments and constructive suggestions on our work. We have addressed all the concerns point-by-point as below.

1) Line 57: Use of the work "hysteretic" is not correct. Consider "limited"

Response #2: As suggested, we have replaced the “hysteretic” with “limited” in the revised manuscript.

2) The authors should reference some of the original articles by David Draper on the role of Mg²⁺ in driving formation of the tertiary structure of RNA. For example, this is consistent with the observation in the present study "that Mg²⁺ has a minimal impact on the folding of individual stems I and II, but is required 126 for the formation of stem IIA/B."

Response #3: As suggested, we have cited the relevant papers by David Draper in the revised manuscript. 1) Page 12, “Numerous studies have shown that Mg²⁺ promotes

and stabilizes the formation of pseudoknot”; 2) Page 12, “Addition of Mg^{2+} prominently increases the fraction of high-FRET state from 23% to 52%, consistent with a stabilizing role of Mg^{2+} in pseudoknot formation”; 3) Page 30, “ Mg^{2+} is essential for neutralizing the negative charge of phosphate moieties in the RNA backbone, which induces structural changes that bring together two RNA regions”.

3) It's not clear if the binding affinity to tRNA of all the modified T-box RNAs used for the FRET analysis were tested and reported. Please include this data.

Response #4: Thanks for the reviewer’s suggestions. We have performed Electrophoretic Mobility Shift Assay (EMSA) for all of the fluorophore-labeled T99 constructs and tested their binding with tRNA, which are summarized in **Fig. S3**. They exhibited slightly weakened binding affinity to tRNA compared with unlabeled T99. In addition, we also conducted smFRET experiments for T99/6-54 in the presence of varying concentrations of tRNA (**Fig. S6**). As can be seen from the FRET histograms, the occupation of middle-FRET increases while low-FRET decreases with the increasing of tRNA concentration, indicating that the middle-FRET corresponds to the tRNA-bound favored conformation. The estimated binding affinity K_d from smFRET experiments was about 40 nM, comparable to that of unlabeled T99 measured by ITC assay. These data indicated that the UBPs-based fluorophore labeling has a minor effect on the binding affinity between T99 and tRNA and can be used in the smFRET experiments.

4) For easy reference it would be helpful to report the concentration of the RNA species used in the experiments be included in the figure legends.

Response #5: As suggested, we have included the RNA concentration in the figure legends for **Fig. 1**.

5) Structures associated with Figures, especially Figure 7, need to be included in the supporting information to allow readers to analyze the structures in detail. Structures

used to initiate the MD simulations should also be supplied to allow others to more readily reproduce the calculations.

Response #6: As suggested, we have added the best-matching structures in **Supplementary Fig. S8**. These best-fitting structures along with the starting structural model for MD simulations were provided in the **Supplementary Data**.

6) Line 526: "incompact" to "less compact"

Response #7: As suggested, we have revised it in the manuscript.

7) SI: The description of the computational methods for the RNA equilibration and production simulations needs to be significantly improved to allow other users to repeat the calculations. Details of the restraints used, atom truncation etc. are required.

Response #8: Thanks for the reviewer's suggestions, we have revised the computational methods in the **Supplementary information extended section 2**.

Response to reviewer 2

Referee #2:

This very thorough study of the folding pathway of the T-box riboswitch shows presents a model of how Mg²⁺-dependent assembly of the aptamer domain of the riboswitch leads to binding of the tRNA anti-codon. The results also show that tRNA binding induces another change in the riboswitch conformation. Thus, as is common in regulatory RNA elements, ligand recognition occurs in multiple stages.

The strength of this study is the combination of SAXS, smFRET, and some MD modeling, which together provide a clear picture of how the RNA structure changes in response to Mg²⁺ and tRNA binding. Another outstanding aspect of this study is the use of non-standard bases to incorporate the desired fluorophores at different sites within the riboswitch. This strategy allowed the authors to test four different labeling

schemes as well as several mutations, resulting in an unusually large amount of smFRET data. The data are high quality, and apart from some technical points noted below, the results are presented completely and clearly.

The weakness of the study design is that the authors only investigated folding of the aptamer domain, and not the full-length riboswitch. This is a shame because recognition of the tRNA acceptor is at the heart of the regulatory mechanism. In addition, the concentration of tRNA wasn't varied, making it more difficult to assess how the T-box responds to its ligand.

Overall, this study adds to the examples of riboswitch folding pathways. The details of the folding pathway of this translational T-box riboswitch are likely to be useful for those studying these riboswitches. The concepts of folding and riboswitch organization are established, however, so in that sense, this study doesn't break new ground.

Response #9: We appreciate the reviewer's positive comments and constructive suggestions on our work and have addressed all the major and minor concerns point-by-point as below.

Specific comments:

1. The ITC results on the tRNA binding in Fig. 1g are very unusual, and demand more explanation. The non-monotonic variation in ΔH for tRNA binding in different Mg^{2+} concentrations indicates a large change in the tRNA binding pathway at high and low Mg^{2+} . This change could be related to preorganization of the aptamer as outlined in Fig. 7 but the authors don't comment on these data at all. The titration with magnesium is not fine enough to know whether the curve at 2 mM Mg^{2+} is a fluke, or a complex trend. Alternatively, the ITC results can be taken out of this manuscript, since they add very little.

Response #10: Thanks for the reviewer's suggestions and we have discussed the ITC results in the revised manuscript (Page 7, Page 27). As the reviewer mentioned, the ΔH for tRNA binding to T99 is relatively small in Mg^{2+} concentrations below 1 mM

probably due to the poor binding, and then becomes larger in 2 mM Mg^{2+} . As the Mg^{2+} concentration increases to 5 mM or 10 mM, ΔH decreases to a smaller value than that in 2 mM Mg^{2+} . In combination with the smFRET data (**Fig. 6**), we speculated that T99 populates an unfolded conformation below 2 mM Mg^{2+} or a preorganized conformation in higher Mg^{2+} (5 mM or 10 mM) and subsequent tRNA binding then induced the folding of T99 into a docked conformation. The transition from unfolded to docked conformation probably results in large structural rearrangement in medium Mg^{2+} (~ 2 mM), accompanying with a larger ΔH than that from preorganized to docked conformation in higher Mg^{2+} .

2. Although the smFRET data in Fig. 3 and Fig. 4 appear high quality and are nicely presented, the unliganded T99 RNA does not show any transitions in the few example trajectories (except when tRNA is added). Yet, the authors describe the RNA structures as dynamic, and the population histograms in Fig. 3b (for example) show more than one FRET state. Do these low FRET states arise from heterogeneity among the RNA molecules? The authors should include more examples in the SI and describe the origins of these FRET peaks more clearly.

Response #11: Thanks for the reviewer's suggestions. We have checked the smFRET data for T99/1-14 in 0 or 20 mM Mg^{2+} carefully and selected some additional representative traces in the **Fig. S4**. In 0 mM Mg^{2+} , T99/1-14 transits between middle- (~0.45) and high-FRET (~0.7). However, in 20 mM Mg^{2+} , T99/1-14 mainly stay in high-FRET and transits to middle-FRET occasionally.

3. Related to the point above, the helices are cartooned as fluctuating between different orientations in Fig. 4f. I find this plausible. However, the basis for this conclusion is not well explained or justified. Do the authors assume that these fluctuations are faster than 0.1 s, and thus broaden the FRET distribution but are not resolved in the movies? Or are they very slow, and thus rarely happen to a given RNA within the span of 20-80 s? These alternatives make a difference to the model for

tRNA recognition.

This point can be addressed by collecting a few longer movies. If photobleaching cannot be reduced, one can acquire data for a long period (5-10 min) by shutting off the excitation between recording intervals. Another way to address it is to jump the magnesium concentration between high and low values while recording, to see how rapidly the RNA population responds. If the response is immediate, it says that the conformational equilibrium is in fast exchange relative to the imaging rate. If some molecules respond and others don't, it suggests a fraction are trapped in a non-productive structure.

Response #12: We appreciate the reviewer's comments and suggestions. We are sorry for the confusions and have corrected the **Fig. 4f** in the revised manuscript. According to our smFRET experiments in **Fig. 3-4**, it's speculated that T99 RNA exhibits three major conformations. 1) In low Mg^{2+} , stem IIA/B is largely unfolded, stem I locates far away from stem II and tends to coaxially stacks with stem II; 2) High Mg^{2+} stabilized the folding of stem IIA/B pseudoknot and its stacking on stem II, which then further promote the directional motion of stem I toward stem II to form a pre-docked conformation. This is different from the stem I/II docked conformation observed in the crystal structure; 3) Subsequent tRNA binding induced the docking of stem I on stem II, and concomitantly, moving away from stem IIA/B.

4. A description of how the RNA quality was validated is missing. What proportion of molecules contained both fluorophores? Does T7 RNAP sometimes mis-incorporate opposite the non-standard template base, or does it just stall? How was the RNA annealing protocol evaluated – for example, is the K_d for tRNA binding comparable to literature values? Can the T99 RNA be saturated with tRNA (>90%)? Does the full-length riboswitch prepared in this way respond to tRNA concentration in the expected range?

Response #13: Thanks for the reviewer's suggestions. We have described the RNA quality validation methods in detail in the **Method section** of the revised manuscript

(Page 36). The labeling efficiency was calculated by measuring the absorptions of T99 RNAs at 260 nm, 546 nm and 646 nm using Nanodrop, in which the labeling efficiencies for Cy3 and Cy5 were ~70% and ~90%, respectively. The NaM-TPT3 unnatural base pairs have been reported to be efficiently recognized by T7 RNAP and incorporated into the RNA transcript, the transcription fidelity in different sequence context is about 90%-100% (PMID: 33118814). In addition, the concentration of TPT3 and NaM used *in vitro* transcription was 0.5 mM to further avoid the mis-incorporation of natural nucleotide against the non-standard template base.

We performed Electrophoretic Mobility Shift Assay (EMSA) for all of the fluorophore-labeled T99 constructs to test their binding with tRNA, which are summarized in **Fig. S3**. In addition, we also conducted smFRET experiments for T99/6-54 in the presence of different concentrations of tRNA in 20 mM Mg²⁺ (**Fig. S6**). As can be seen from the FRET histograms, the occupation of middle-FRET increases while low-FRET decreases with the increasing of tRNA concentration, indicating that the middle-FRET corresponds to the tRNA-bound favored conformation. The estimated binding affinity K_d was about 40 nM, comparable to that of unlabeled T99 measured by ITC assay. These data indicated that the UBP-based fluorophore labeling has a minor effect on the binding affinity between T99 and tRNA and can be used in the smFRET assay. Consequently, the T99 RNA can be saturated by the presence of a large excess of tRNA.

For the full-length T-box riboswitch containing both of aptamer domain and discriminator domain, we have used the similar strategy to prepare the dye-labeled construct for smFRET assay and they also exhibited comparable binding affinity to tRNA as unlabeled sample. These data might be reported in our future work about the full-length T-box riboswitch.

5. The Mg titrations in Fig. 6a,b are really beautiful and clearly show that tRNA binding induces a new conformational state of the riboswitch. On line 391, line 423 and in Figure 6, however, the authors mention that they have measured the tRNA

binding kinetics. Do the authors assume that the high FRET state Fig. 6c represents tRNA binding? How do they distinguish binding from a conformational change in the tRNA-riboswitch complex? Please explain in the manuscript. Also, without varying the tRNA concentration, it is hard to estimate the on rate – this caveat should be also noted in the main text.

Response #14: Thanks for reviewer’s comments and suggestions. It’s indeed difficult to distinguish tRNA binding from a conformational change in the tRNA-riboswitch complex in our smFRET assay without labeling tRNA. We have corrected the statement about “Mg²⁺-dependent tRNA binding kinetics” as “Mg²⁺-dependence of tRNA-induced conformational changes of T99” in the revised manuscript.

We have conducted smFRET experiments for T99/6-54 in the presence of varying concentrations of tRNA (**Fig. S6**). As can be seen from the FRET histograms, the occupation of middle-FRET increases while low-to-middle FRET decreases with the increasing of tRNA concentration (1 nM-2 μM), indicating that the low-to-middle- and middle-FRET corresponds to the apo- and tRNA-bound favored conformation, respectively. Transition rates k_{dock} and k_{undock} for T99/6-54 in different tRNA concentrations were calculated and plotted against tRNA concentrations. As tRNA concentration increases, k_{dock} increases significantly while k_{undock} decreases mildly especially in 10 nM to 2 μM, indicating that the recognition between T99 and tRNA favored an “induced-fit” mechanism.

6. As I understand it, computed RNA models were selected that fit the SAXS data – was this done for an ensemble, or individual structures? Can the experimental curves be fit well by a single structure? Please explain this more clearly in the main text and methods.

Response #15: Thanks for reviewer’s suggestions, we have revised this part in the main text and methods (**Supplementary extended section 2**). For each RNA, we generate conformational pool and screen out the structural models using the respective SAXS data. Almost in all cases, the theoretical scattering curve of the

selected single structure can fit well with the experimental scattering curves ($\chi^2 < 1.5$), which share high structural similarity (**Supplementary Fig. S8**).

7. Fig. 7 superimposes individual ribbons on the envelopes, which I presume come from bead models (DAMMIN). First, if more than one structure can fit the data, then this should be shown in Fig. 7 in some fashion. Second, the bead models are not the best way of interpreting the SAXS data, as the authors likely know. DAMMIN is a particularly poor choice for RNA that is more electron dense and that has different patterns of solvation and ion association than proteins. Did the authors try other software packages? Or, perhaps there is no need to show the bead models if they weren't used to evaluate the calculated RNA structures.

Response #16: Thanks for the reviewer's comments and suggestions. Indeed, there exist other structural models which can fit the SAXS data. However, they share highly structural similarity, thus they can be classified into one group. For clarity, we superimposed these structures in **Fig. S8** instead of in **Fig. 7**. The envelopes were indeed derived from bead models and just used to show the consistency of the selected structural models with SAXS data. We could remove the bead models if necessary.

8. More description of the data analysis and its errors is needed – in addition to the number of molecules analyzed, the authors should also provide some estimate of the errors in the HMM fitting, the proportion of molecules in each FOV that were used. For the population histograms, it would be helpful to know how many frames of each movie were used.

Response #17: Thanks for reviewer's suggestions and we have described the smFRET data analysis and errors estimate in detail in the **Methods section** of the revised manuscript (Page 36-37). We want to point out that the fitting errors after HMM modeling are small, therefore, the experimental errors among different repeats were plotted in our figures.

9. *The introduction is well written but quite long (as is the discussion). To help readers appreciate their work, the authors may want to consider saving some of this background material for a review article, and instead focus on the translational T-box riboswitch they have studied. It is not until line 116 that one learns that folding of a translational T-box (vs transcriptional T-box) is the new question here!*

Response #18: Thanks for reviewer's comments and suggestions. We have refined the introduction to make it more concise and become more readable to readers in the revised manuscript.

10. *To aid understanding, please avoid acronyms, and if they must be used, be sure to define them in the main text. I found the acronym "RRI" particularly unhelpful and unnecessary.*

Response #19: Thanks for the reviewer's suggestions. We have revised it in the manuscript and replaced the "RRI" with "RNA-RNA interactions".

11. *A minor suggestion is to move the SAXS data in Figure 1 to Figure 7, where the data will be used to model the folding intermediates. Figure 1 is quite complicated, yet the results here were almost not used. The real focus of this paper is on the smFRET results and it would help to get to that immediately.*

Response #20: Thanks for the reviewer's nice suggestions. The SAXS data in **Fig. 1d-f** were used to elucidate the global conformational changes of T99 induced by Mg^{2+} and tRNA, which inspires us to probe the detailed inter-domain or intra-domain conformational dynamics of T99 by smFRET assay. We have tried to move the SAXS data in **Fig. 1** to **Fig. 7**, however, we found that placing the SAXS data in **Fig. 1** seems to be more logical for the overall story. We then decided to remain the SAXS data still in **Fig. 1**.

12. In the legend to Fig. 3 (and following), please explain that the blue lines represent Cy5 intensity and green lines represent Cy3 intensity.

Response #21: As suggested, the legend to **Fig. 3-6** have been revised in the manuscript.

13. The free energy diagrams in Fig. 6m,n show a main conclusion of the paper, yet this is almost buried in the amount of detail. I suggest moving the data for the mutants in Fig. g-l to the SI, so that more space and prominence can be given to the results that count.

Response #22: Thanks for the reviewer's comments and suggestions. The data for the truncation mutants (T77 and T89) in **Fig. 6g-l** (old version, **6h-o** in revised version) in different Mg^{2+} concentrations were used to assess the role of the structural integrity of stem IIA/B in the folding and tRNA recognition of T99. For easier comparison with T99, we think it is more appropriate to place the data for T89 and T77 still in **Fig. 6**.

Response to reviewer 3

Referee #3:

The manuscript by Niu et al. utilizes a thorough combination of SAXS, smFRET, and molecular dynamics simulations to investigate tRNA decoding by the T-box riboregulator. T-boxes represent unique genetic regulatory elements that have analogous function to riboswitches but bind non-acylated tRNA instead of small molecule ligands. By decoding and sensing the aminoacylation status of the bound tRNA, T-boxes facilitate regulatory responses to the depletion of specific amino acids. Here, the authors examine decoding with a minimal decoding module consisting of Stem-I, Stem-II, and Stem-IIA/B domains. A major strength of the study is the development and use of the unnatural base pair (UBP) system for multiple dual labeling schemes of the decoding module. This allowed them to survey using smFRET the relative motions of all domains within the module across Mg^{2+} concentrations and in the presence of tRNA. An important conclusion from this approach is that

pseudoknot formation is critical for tRNA binding, presumably by sterically restricting the motion of Stem-I, thereby encouraging interaction between the specifier and Stem-II S-loop. Overall, the data within the manuscript are a very nice complement to recent high-resolution T-box structures and single-molecule experiments.

Response #23: We appreciate the reviewer's positive comments and constructive suggestions on our work and have addressed all the concerns point-by-point as below.

However, in some cases the authors' conclusions could be presented more consistently and clearly. Questions and comments are below:

1. Something that bears mentioning in the manuscript is that in all experiments the decoding module is being studied at equilibrium. In the cell, these riboregulators function co-transcriptionally and likely sample conformations that cannot be captured in these experiments. For example, the tRNA will be able to interact with Stem-I specifier shortly after it is transcribed and before Stem-II or Stem-IIA/B are made. It would be very worthwhile for the authors to make mention of these caveats in the discussion and possibly frame some of their conclusions in light of how decoding may function co-transcriptionally. Framing the role of the Stem-IIA/B pseudoknot in a co-transcriptional scenario is especially important.

Response #24: We appreciate the reviewer's constructive suggestions. We have discussed the co-transcriptional folding and tRNA recognition mechanism of T-box riboswitch in the **Discussion** section of the revised manuscript (Page 28-29).

2. The prevailing model of riboswitch-ligand interaction is that the RNA can alternate between the ligand-bound and apo states even when the ligand is not present. Typically, these two states differ in their base-pairing scheme, and alternating between these two states tends to be rate-limiting, therefore the mechanism typically involves ligand-induced conformation capture of the correctly base-paired conformer. Here the authors propose an "induced-fit" model to define the action of the tRNA decoding module of T-box, where no alternative base-pairing is involved. The authors

need to clearly define the importance of this induced-fit action in the context of the entire riboswitch to avoid confusing the readers that riboswitches may function entirely through an induced-fit mechanism.

Response #25: As reviewer mentioned, riboswitches recognize their ligands mainly through “conformational selection” or “induced-fit” mechanism and our smFRET data indicated that T99 probably actions through the latter. We have discussed the ligand recognition mechanism by T-box in the context of the entire riboswitch in the revised manuscript (Page 29-30).

3. The authors mainly use the secondary structure model to describe their induced-fit model. Given that high-resolution T-box/tRNA structures are available, it is more effective to use structural models to describe their mechanism. This reviewer encourages the authors to generate a morphing movie using Pymol or Chimera to describe their envisioned conformational changes. The authors could highlight the positions of the fluorophores on the model and their distances. Make sure to clearly define which state is hypothetical, which state is based on real structures.

Response #26: As suggested, we have generated movies to describe the conformation changes of T99 RNA induced by Mg^{2+} and tRNA binding (**Supplementary Movie**). Overall, the structure models in **Fig. S8** agree well with our SAXS data and exhibit highly structural similarity. Meanwhile, the inter-nucleotides distance (6-54) for different constructs at different conditions (Mg^{2+} , tRNA) also reflect a trend consistent with our FRET observables. Therefore, the representative structural models (top 1 of each group) presented in **Fig.7** should be regarded as “real structure” that are at least consistent with our SAXS and FRET data.

4. Along the same line, is there any Mg^{2+} binding sites in the structure that could explain the higher FRET state in T-box in the absence of tRNA? For example, there is a mg bound by G41/G42 in the published structure. Is this G-C pair conserved for the purpose of stabilizing the high FRET state? What happens when the G-C pairs here

are changed to A-U pairs?

Response #27: Thanks for the reviewer's comments and suggestions. No Mg^{2+} is found in the crystal structure of *N. farcinica* T-box/tRNA complex (PDB: 6UFM) probably due to the limited resolution. However, one Mg^{2+} is found to be located in the stem IIA/B pseudoknot of the crystal structure of *M. tuberculosis* T-box/tRNA complex (PDB: 6UFG) and one Mg^{2+} is found to be located in the stem II (G41/G42) region of the crystal structure of *M. tuberculosis* T-box/tRNA-2',3'cp complex (PDB: 6UFH) as reviewer mentioned.

The observed stabilization effect of stem IIA/B pseudoknot by Mg^{2+} could be well explained by the Mg^{2+} binding site near the stem IIA/B region (**Fig. 3d-e**). According to our data, Mg^{2+} -stabilized stem IIA/B pseudoknot tends to coaxially stack with stem II, and leads to the movement of stem I toward stem II in high Mg^{2+} in the absence of tRNA (higher FRET for T99/6-54 and T99/14-54 in high Mg^{2+}).

The Mg^{2+} bound by G41/42 pairs is located in the duplex region of stem II and doesn't involved in any tertiary interactions, so it's speculated that the Mg^{2+} -bound G-C base pairs may have a minor effect on the Mg^{2+} -dependence of T99 RNA's folding.

5. The effect of the magnesium concentration should be further discussed. Stabilization of the tRNA/T-box Stem-I specifier and Stem-II S-loop requires 10 mM Mg^{2+} . However, the free $[Mg^{2+}]$ is typically reported in the 1-5 mM range, therefore the complex is expected to be less stable in vivo. The functional implications should be discussed.

Response #28: Thanks for reviewer's comments and suggestions. As reviewer mentioned, the near-physiological Mg^{2+} is only about 1-5 mM, which is not enough for the stable binding of tRNA by T77 (T-box stem-I and stem II) or T89. However, the cellular environment is more crowded than *in vitro* solution buffer and contains many RNA folding chaperones, which will promote the RNA folding and thus reduce the requirement for high Mg^{2+} concentration *in vivo*. We have discussed this in the

revised manuscript (Page 22).

6. *The authors make two seemingly contradictory statements within the manuscript:*

1). *“Taken together, these smFRET data reveal a sequential docking mechanism for T99 that high Mg²⁺ facilitates the folding of stem IIA/B pseudoknot and its stacking on stem II for pre-docking of stems I and II to form a competent tRNA binding conformation, and subsequent tRNA binding drives further docking of stem I on stem II, and concomitantly, moving away from stem IIA/B.”*

2). *“It’s likely that the initial contact between the tRNA anticodon and Specifier in stem I drives the formation of a helical, stacked conformation for the specifier and then induces the docking of stem I towards stem II via backbone interactions with the S-turn region, which in turn reinforces the Specifier-anticodon interactions.”*

In statement 1, Stem-I and Stem-II preform a tRNA binding site while in statement 2, the tRNA binds Stem-I first before docking with Stem-II. The confusion between these two statements stems from the ambiguous definition of “pre-docking” in the manuscript. Does pre-docking refer to a preformed Specifier-S-turn binding cleft for tRNA? Does the low FRET state in high Mg²⁺ correspond to the pre-docked conformation? Why does this low FRET state still exist in the S-turn mutants? Clearing up these questions would greatly improve the clarity of the manuscript and models presented therein.

Response #29: We are sorry for this confusion. We defined the low-to-middle-FRET for T99/6-54 in high Mg²⁺ as “pre-docked” state, but it just refers to the conformational state in which stem I/II being close to each other, instead of a preformed Specifier-S-turn binding cleft for tRNA. In that case, the low-to-middle FRET state still exists in the S-turn mutants even in high Mg²⁺ because the Specifier-S-turn interaction didn’t form before tRNA anticodon binding. We have revised the description about “pre-docked” in the revised manuscript (Page 14).

6. *The figure of the 3D structure showing dye placements for the different constructs*

in the FRET experiments (Figure 2D) is difficult to interpret. A better representation would be to remove the dashed distance markers and simply label the different dye sites. The distances between the different dye pairs could be provided in a separate table.

Response #30: Thanks for the reviewer's suggestions. As suggested, we have removed the dashed distance markers and summarized the distances between different dye pairs in a table next to the 3D structure of T99 in **Fig. 2d**.

8. The authors should consider changing "hysteretic" in the sentence "However, our understanding of how RRI occurs and drive the RNA folding and conformational dynamics remains relatively hysteretic." to something like "lacking" or "minimal".

Response #31: Thanks for the reviewer's suggestions. We have replaced "hysteretic" with "limited" in the revised manuscript.

Response to reviewer 4

Referee #4:

*In the manuscript entitled "Structural and dynamic mechanisms for coupled folding and tRNA recognition of a translational T-box riboswitch" the authors present an analysis of the translational T-box regulator from *N. farcinica*. This is the first such study of which I am aware of a translational T-box, there are existing smFRET studies of the more prevalent transcriptional T-boxes (Suddala Nat. Com 2018, Zhang et al. eLife 2018). This work includes what appears to be some nice smFRET that is enabled by the incorporation of non-natural bases to allow the integration of dyes at specific sites. The method for incorporation looks to be quite flexible and is likely of use to many researchers, although it does look to have been previously published by same group (Wang et al. PNAS 2020). Not much time in the discussion is spent on the biological significance of the findings, and in particular how transcriptional and translational T-boxes may have similar distinct mechanisms for tRNA recognition. Overall, this is a nice study that adds knowledge to the growing cannon on T-box*

recognition, but ultimately does not fully contextualize its findings.

Response #32: We thank the reviewer for the positive comments on our manuscript and address the concerns point-by-point as below.

Major comments:

20 mM Mg²⁺ is high and not physiologically relevant. Physiological Mg²⁺ is thought to be approximately 0.5 to 2 mM, and even the authors own data suggest that the Mg²⁺ concentrations in the range of 2-5 mM are sufficient for tRNA binding. It is not clear why the data in Figures 3 and 4 were collected at such a high concentration of Mg²⁺. It may be that results 20 mM looks pretty much the same as lower concentrations (data on Fig. 4b (20 mM Mg²⁺) and 5b (7.5 mM Mg²⁺) look similar to each other), but it is clear that Mg²⁺ does have an impact on the folding as monitored by smFRET (Fig. 6ab), yet the authors capture only a non-physiological snapshot for much of the work conducted.

Response #32: Thanks for the reviewer's comments. In our manuscript, we performed smFRET experiments for T99 RNA in high Mg²⁺ for the following reasons. Though the physiological Mg²⁺ is only about 0.5 to 2 mM, the *in vivo* cellular environment is quite crowded than the *in vitro* aqueous buffer and contains many RNA folding chaperons, which also contributes greatly to the proper folding of RNAs (PMID: 27378777) and thus reduce the requirement for high Mg²⁺ concentrations *in vivo*. As Mg²⁺ has been shown to be essential for the structure and tertiary folding of RNAs, so we increase Mg²⁺ to a higher concentration in smFRET experiments to stabilize the folding of T99 RNA, which has been extensively used in previous biophysical studies (PMID: 28825710, 28920931). These *in vitro* results may not apply directly to the cellular conditions in bacteria, but still shed insights into the physiological roles of Mg²⁺ to T-box's folding and tRNA recognition functions. Our smFRET data indicated that Mg²⁺ not only promote the folding of T99 but also promote the binding between T99 and tRNA.

The authors have not substantiated this line from the discussion (pg 26 line 517). “To achieve functional conformations, the nascent RNA in general must proceed through a folding pathway, which could initially yield a large pool of partially folded conformations and nonfunctional states”. The authors assessed two truncations, and demonstrate that one of them (T77) is not competent to bind. This does not constitute characterization of the folding pathway for the RNA.

Response #33: Thanks for reviewer’s comments. We have revised this sentence and also cited relevant study about this statement (Page 27).

The biological impact of this work is missing from the text. What do we know now about mechanism, biology, or drugability, that we did not know previously? This work suggests that many of the findings are similar to those previously published (Suddala Nat. Com 2018, Zhang et al. eLife 2018), although there do seem some distinctions between this potentially concerted mechanism and the two-step mechanism described in prior works. The lack of contextualization decreases the general interest in the work substantially.

Response #34: Thanks for the reviewer’s comments and suggestions about the discussion section. The biological importance of our work includes: 1) Our comprehensive dynamic analysis unravels an “induced-fit” recognition mechanism between T-box and tRNA, which is drastically different from a pre-organized tRNA binding groove and a “lock-and-key” binding model suggested in the snapshots of previous crystal structures. 2) Data on different truncation mutants suggest that tRNA decoding by *ileS* T-box occurs in a co-transcriptional manner and stem IIA/B pseudoknot delicately coordinates the folding of T99 during transcription for efficient tRNA recognition. In addition, tRNA binding by *ileS* T-box at the early stage of transcription is clearly different from that in *glyQS* T-box, which only begins to recognize tRNA anticodon until stem I is almost fully transcribed. The early ligand binding during transcription is presumably related to the timely downstream gene expression regulation and awaits further investigations. 3) The structural dynamics of

T-box in response to Mg²⁺ and tRNA binding, implying that targeting the dynamic conformational ensemble of T99 provides a potential and promising avenue for the design and development of RNA-targeted novel antibiotics. We have revised the discussions in the revised manuscript.

Minor comments:

*The genus name of *N. farcinica* is never given in the main text.*

Response #35: Thanks for reviewer's suggestions. We give its full name where it is first mentioned in the revised manuscript (Page 5).

The use of both T99 and WT is a bit confusing. I think these are the same construct, but both terms are used at different points.

Response #36: We feel sorry about this confusion. T99 and WT are indeed the same construct. We have replaced "WT" with "T99" in the revised manuscript.

A better indication on the cartoons in Figure 5 of the relationship between the fluorophores and the base changes would be nice. The U90C is indicated fairly clearly, but the G85C is not similarly indicated.

Response #37: As suggested, we have revised the cartoon representation for G85C/6-54 in **Fig. 5a**.

Figure 6 is difficult to interpret due to inadequate labelling with the figure and in accurate legend. Parts d-i correspond to different things in a 3x3 grid which is labelled d-f in the top row, g-i for the second row and j-l for the third row. However, the legend refers to "(d-j)", "(e-k)", and "(f-l)". These are not accurate. Also, as the illustration itself is not labelled itself, it is difficult to understand which constructs are represented in which figure part.

Response #38: We feel sorry about the confusion. We have reorganized and labeled

the **Fig. 6** and revised the figure legend.

REVIEWERS' COMMENTS

Reviewer #1 (Remarks to the Author):

This is an outstanding study. The authors have addressed all of my concerns.

Reviewer #2 (Remarks to the Author):

The authors have addressed many of the substantive comments of the reviewers in their revisions. Although the biological relevance of the experiments at high Mg²⁺ and in the absence of transcription is still limited, the data provide new information on the path of tRNA recognition by this riboswitch. The addition of smFRET at different tRNA concentrations (Fig. S6) substantially supports the conclusion that the T-box riboswitch binds the tRNA via an induced fit mechanism and is a noted improvement. The authors have also clarified some parts of their manuscript which were confusing before. Overall, the revised manuscript is much improved.

There are a few small remaining points that could benefit from further correction:

1. The added EMSA tRNA binding assays in Fig. S3 show most of the fluorophore labels mildly destabilize tRNA binding, but are not hugely perturbing. The exception is the 35-54 labeling combination, which changes the riboswitch gel mobility, and decreases tRNA binding ~10 fold. Since this labeled RNA is not used to support the main conclusions of the paper, it is fine. However, the authors should take out the conclusion on pg. 12, line 233 that stem II is rigid – I think they cannot conclude anything from this particular labeled RNA.
2. The text on pg. 11 line 217 still claims that T99/1-14 samples different FRET states under all conditions, but this is not shown in the main figure, only the new Fig. S4 (response #11). Ideally, it should be in the main figure.
3. The explanation of how the SAXS data were modeled is much improved, including the superposition of top models in Fig. S8. As the bead models weren't actually used, and don't represent the agreement between the model and the data, they should be removed from Fig. 7. Otherwise, the figure just gives a false idea of what was done that can mislead readers less familiar with the method (response #16).

4. Typos:

Pg. 21, line 424, “occupancy” rather than “occupation”.

Pg. 21, line 425, middle-FRET increases (not decreases).

Fig. S6b, unocking should be undocking.

Reviewer #3 (Remarks to the Author):

The authors have addressed all of my concerns. Their responses to the other reviewers are satisfactory to me. I therefore recommend the publication of this work without further delay.

Reviewer #4 (Remarks to the Author):

The authors have addressed my concerns.

Response to reviewer 2

Referee #2:

The authors have addressed many of the substantive comments of the reviewers in their revisions. Although the biological relevance of the experiments at high Mg²⁺ and in the absence of transcription is still limited, the data provide new information on the path of tRNA recognition by this riboswitch. The addition of smFRET at different tRNA concentrations (Fig. S6) substantially supports the conclusion that the T-box riboswitch binds the tRNA via an induced fit mechanism and is a noted improvement. The authors have also clarified some parts of their manuscript which were confusing before. Overall, the revised manuscript is much improved.

There are a few small remaining points that could benefit from further correction:

Response #1: We thank the reviewer 2 for the positive comments and constructive suggestions on our work. We have addressed all the concerns point-by-point as below.

1) The added EMSA tRNA binding assays in Fig. S3 show most of the fluorophore labels mildly destabilize tRNA binding, but are not hugely perturbing. The exception is the 35-54 labeling combination, which changes the riboswitch gel mobility, and decreases tRNA binding ~10 fold. Since this labeled RNA is not used to support the main conclusions of the paper, it is fine. However, the authors should take out the conclusion on pg. 12, line 233 that stem II is rigid – I think they cannot conclude anything from this particular labeled RNA.

Response #2: Thanks for reviewer's suggestions. We have removed the sentence "These results suggest that stem II is relatively rigid and its folding is less affected by Mg²⁺ or tRNA binding." in the revised manuscript.

2) The text on pg. 11 line 217 still claims that T99/1-14 samples different FRET states under all conditions, but this is not shown in the main figure, only the new Fig. S4 (response #11). Ideally, it should be in the main figure.

Response #3: As suggested, we have replaced the smFRET traces in **Fig. 3b** with new smFRET traces showing transitions in different FRET states in the revised manuscript.

3) The explanation of how the SAXS data were modeled is much improved, including the superposition of top models in Fig. S8. As the bead models weren't actually used, and don't represent the agreement between the model and the data, they should be removed from Fig. 7. Otherwise, the figure just gives a false idea of what was done that can mislead readers less familiar with the method (response #16).

Response #4: Thanks for the reviewer's suggestions. We have removed the beads model in **Fig. 7a** in the revised manuscript.

4) Typos:

Pg. 21, line 424, "occupancy" rather than "occupation".

Pg. 21, line 425, middle-FRET increases (not decreases).

Fig. S6b, unocking should be undocking.

Response #5: Thanks for reviewer's careful inspections. As suggested, we have corrected these writing mistakes in the revised manuscript.